# NEURAL VARIATIONAL DROPOUT PROCESSES

**Insu Jeon**[1]**, Youngjin Park**[2]**, Gunhee Kim**[1]
[1]Seoul National University; [2]Everdoubling LLC., Seoul, South Korea
`insuj3on@gmail.com, youngjin@everdoubling.ai, gunhee@snu.ac.kr`

## ABSTRACT

Learning to infer the conditional posterior model is a key step for robust meta-learning. This paper presents a new Bayesian meta-learning approach called Neural Variational Dropout Processes (NVDPs). NVDPs model the conditional posterior distribution based on a task-specific dropout; a low-rank product of Bernoulli experts meta-model is utilized for a memory-efficient mapping of dropout rates from a few observed contexts. It allows for a quick reconfiguration of a globally learned and shared neural network for new tasks in multi-task few-shot learning. In addition, NVDPs utilize a novel prior conditioned on the whole task data to optimize the conditional *dropout* posterior in the amortized variational inference. Surprisingly, this enables the robust approximation of task-specific dropout rates that can deal with a wide range of functional ambiguities and uncertainties. We compared the proposed method with other meta-learning approaches in the few-shot learning tasks such as 1D stochastic regression, image inpainting, and classification. The results show the excellent performance of NVDPs.

## 1 INTRODUCTION

In traditional machine learning, a large amount of labeled data is required to train deep models (LeCun et al., 2015). In practice, however, there are many cases where it is impossible to collect sufficient data for a given task. On the other hand, humans can quickly understand and solve a new task even from a few examples (Schmidhuber, 1987; Andrychowicz et al., 2016). This distinguishing characteristic of humans is referred to as the meta-learning ability, which enables them to accumulate past learning experiences into general knowledge and to utilize it for efficient learning in the future. Incorporating the meta-learning capability into artificial machines has long been an active research topic of machine learning (Naik & Mammone, 1992; Vinyals et al., 2016; Snell et al., 2017; Ravi & Larochelle, 2017; Finn et al., 2017; Lee et al., 2019c; Hospedales et al., 2021).

Recently, Bayesian meta-learning methods have been attracting considerable interest due to incorporating uncertainty quantification of the Bayesian framework into the efficient model adaptation of meta-learning approaches. An earlier study (Grant et al., 2018) extended the optimization-based deterministic approach, model-agnostic meta-learning (MAML) (Finn et al., 2017), to a hierarchical Bayesian framework (Daumé III, 2009). Later, optimization-based variational inference (Yoon et al., 2018; Finn et al., 2018; Ravi & Beatson, 2019; Lee et al., 2019a; Nguyen et al., 2020) and model-based Bayesian meta-learning methods (Gordon et al., 2018; Garnelo et al., 2018a;b; Iakovleva et al., 2020) were presented. These methods achieved outstanding results in various few-shot[1] regression and classification tasks. In fact, the adaptation of deep models that can estimate uncertainties is a core building block for reliable machine learning systems in real-world applications with high judgmental risks, such as medical AI or autonomous driving systems.

Inspired by recent studies, we propose a new model-based Bayesian meta-learning approach, neural variational dropout processes (NVDPs). The main contribution of this work is to design a new type of Neural Network (NN) based conditional posterior model that can bypass the under-fitting and posterior collapsing of existing approaches (Gordon et al., 2018; Garnelo et al., 2018a;b; Iakovleva et al., 2020). NVDPs extend the simple yet effective posterior modeling of Variational Dropout (VD) (Kingma et al., 2015; Gal & Ghahramani, 2016; Molchanov et al., 2017; Hron et al., 2018;

---

[1]The few-shot learning assumes only a few examples are available for each task, with the number of tasks being large (Lake et al., 2015).

Liu et al., 2019) in the context of meta-learning. A novel low-rank product of Bernoulli experts meta-model is utilized in the *dropout* posterior to learn task-specific dropout rates conditioned on a few learning examples. In this way, a full conditional posterior model over the NN's parameters can be efficiently obtained. In addition, we also propose a new type of prior to optimize the conditional *dropout* posterior in variational inference, which supports the robust training of conditional posteriors. Although it is developed in conjunction with NVDPs, the formulation allows its adoption to other recent methods, and we show the effectiveness. We have evaluated NVDPs compared with other methods on various few-shot learning tasks and datasets. The experiments show that NVDPs can circumvent the under-fitting and posterior collapsing and achieve outstanding performances in terms of log-likelihood, active learning efficiency, prediction accuracy, and generalization.

## 2 BAYESIAN META-LEARNING

### 2.1 AMORTIZED VARIATIONAL INFERENCE FRAMEWORK IN THE MULTI-TASK DATA

A goal in meta-learning is to construct a model that can quickly solve new tasks from small amounts of observed data. To achieve this, it is important to learn a general (or task-invariant) structure from multiple tasks that can be utilized for the efficient model adaptation when necessary. Bayesian meta-learning methods (Gordon et al., 2018; Garnelo et al., 2018b; Ravi & Beatson, 2019) formulate this objective of meta-learning as an amortized variational inference (VI) of the posterior distribution in multi-task environments. Suppose a collection of $T$ related tasks is given, and each $t$-th task has the training data $\mathcal{D}^t$ containing $N$ i.i.d. observed tuples $(\mathbf{x}^t, \mathbf{y}^t) = (x_i^t, y_i^t)_{i=1}^N$. Then, the evidence lower-bound (ELBO) over the log-likelihood of the multi-task dataset can be derived as:

$$\sum_{t=1}^T \log p(\mathcal{D}^t; \theta) \geq \sum_{t=1}^T \{\mathbb{E}_{q(\phi^t|\mathcal{D}^t)}[\log p(\mathbf{y}^t|\mathbf{x}^t, \phi^t)] - \mathrm{KL}(q(\phi^t|\mathcal{D}^t; \theta)||p(\phi^t))\}. \qquad (1)$$

Here, $p(\mathbf{y}^t|\mathbf{x}^t, \phi^t)$ is a likelihood (or NN model) on the $t$-th training data and $\phi^t$ is a $t$-th task-specific variable (*i.e.* a latent representation or weights of NN) and $q(\phi^t|\mathcal{D}^t; \theta)$ is a tractable amortized variational posterior model utilized to approximate the true unknown posterior distribution over $\phi^t$ for each given $t$-th task data (i.e., $p(\phi^t|\mathcal{D}^t)$) (Kingma & Welling, 2013; Gordon et al., 2018; Garnelo et al., 2018b; Ravi & Beatson, 2019; Iakovleva et al., 2020). The parameter $\theta$ represents the common structure that can be efficiently learned across multiple different tasks. The prespecified prior distribution $p(\phi^t)$ in the Kullback–Leibler (KL) divergence term provides a stochastic regularization that can help to capture the task-conditional uncertainty and prevent the collapsing of $q(\phi^t|\mathcal{D}^t; \theta)$. In fact, the maximization of the ELBO, right side of equation 1, with respect to the conditional variational posterior model is equivalent to the minimization of $\sum_{t=1}^T \mathrm{KL}(q(\phi^t|\mathcal{D}^t; \theta)||p(\phi^t|\mathcal{D}^t))$. Essentially, the goal in the amortized VI of Bayesian meta-learning is to learn the inference process of the true conditional posterior distribution via the variational model $q(\phi^t|\mathcal{D}^t; \theta)$ and the shared general structure $\theta$ across multiple tasks since this task-invariant knowledge can later be utilized for the efficient adaptation of the NN function on new unseen tasks. The approximation of the conditional posterior also enables ensemble modeling and uncertainty quantification.

### 2.2 THE EXISTING CONDITIONAL POSTERIOR AND PRIOR MODELINGS

Many of the recent Bayesian meta-learning approaches can be roughly categorized into the optimization-based (Grant et al., 2018; Yoon et al., 2018; Finn et al., 2018; Ravi & Beatson, 2019; Lee et al., 2019a; Nguyen et al., 2020) or the model-based posterior approximation approaches (Gordon et al., 2018; Garnelo et al., 2018a;b; Kim et al., 2019; Iakovleva et al., 2020). In the optimization-based approaches, the task-specific variable $\phi^t$ can be seen as the adapted NN weights. For example, the deterministic model-agnostic meta-learning (MAML) (Finn et al., 2017) can be considered as learning a Dirac delta variational posterior modeling (i.e., $q(\phi^t|\mathcal{D}_C^t; \theta) \approx \delta(\phi^t - \mathrm{SGD}_j(\mathcal{D}_C^t, \theta))$) where the goal is to learn the shared global initialization NN's parameters $\theta$ such that a few $j$ steps of SGD updates on the small subset[2] $\mathcal{D}_C^t$ of the $t$-th dataset $\mathcal{D}^t$ provides a good approximation of the task-specific weights $\phi^t$. Many optimization-based Bayesian meta-learning approaches such

---

[2]The subset $\mathcal{D}_C^t(\subseteq \mathcal{D}^t)$ is known as a context (or support) set for each task. A small $S$ size of context set (e.g., 1-shot, 5-shot, or random) is often used in the few-shot learning tasks (Lake et al., 2015).

as LLAMA (Grant et al., 2018), PLATIPIS (Finn et al., 2018), BMAML (Yoon et al., 2018), and ABML (Ravi & Beatson, 2019) have incorporated the Gaussian type of posterior and prior models into the deterministic adaptation framework of MAML in order to improve the robustness of models. Optimization-based approaches can be applied to various types of few-shot learning tasks, but the adaptation cost at test time is computationally expensive due to the inversion of the Hessian or kernel matrix; they may not be suitable for environments with limited computing resources at test.

On the other hand, the model-based Bayesian meta-learning approaches such as VERSA (Gordon et al., 2018) or Neural Processes (Garnelo et al., 2018a;b) allow an instant estimation of the Bayesian predictive distribution at test time via NN-based conditional posterior modeling. VERSA employs an additional NN-based meta-model $g_\theta(\cdot)$ to directly predict the Gaussian posterior of agent NN's task-specific weight $\phi^t$ from the small context set (*i.e.*, $q(\phi^t|\mathcal{D}_C^t;\theta) = \mathcal{N}(\phi^t|(\mu,\sigma) = g_\theta(\mathcal{D}_C^t)))$. In this case, the shared structure $\theta$ represents the meta-model's parameters. However, the direct approximation of agent NN weights in VERSA could limit their scalability due to the large dimensionality of the NN's weights; they only consider the task-specific weights for the single softmax output layer. In addition, VERSA did not utilize any task-specific prior $p(\phi^t)$, so the conditional posterior could collapse into a deterministic one while training with the Monte Carlo approximation (Iakovleva et al., 2020). Instead of the NN's weights, the Neural Processes (NPs) model the conditional posterior with a latent representation (*i.e.*, $q(z^t|\mathcal{D}^t;\theta) = \mathcal{N}(z^t|(\mu,\sigma) = g_\theta(\mathcal{D}^t)))$ (Garnelo et al., 2018b). Regarding the task-specific latent representation as a second input to the likelihood $p(\mathbf{y}^t|\mathbf{x}^t, z^t)$ (similar to Kingma & Welling (2013) or Edwards & Storkey (2017)) provides another efficient way to transfer the conditional information (Garnelo et al., 2018b). However, the following studies indicate that NPs often suffer from under-fitting (Kim et al., 2019) and posterior-collapsing behavior (Grover et al., 2019; Le et al., 2018). We conjecture that these behaviors are partially inherent to its original variational formulation. Since the variational posterior in NPs rely on the whole task data while training, the posterior do not exhibit good generalization performance at test. In addition, it is well known that the highly flexible NN-based likelihood can neglect the latent representation in variational inference (Hoffman & Johnson, 2016; Sønderby et al., 2016; Kingma et al., 2016; Chen et al., 2017; Yeung et al., 2017; Alemi et al., 2018; Lucas et al., 2019; Dieng et al., 2018); we could also observe the posterior collapsing behavior of the NPs in our experiments.

## 3   Neural Variational Dropout Processes

This section introduces a new model-based Bayesian meta-learning approach called Neural Variational Dropout Processes (NVDPs). Unlike the existing methods such as NPs or VERSA employing conditional latent representation or direct modeling of NN's weights, NVDPs extend the posterior modeling of the Variational Dropout (VD) in the context of meta-learning. We also introduce a new type of task-specific prior to optimize the conditional *dropout* posterior in variational inference.

### 3.1   A Conditional Dropout Posterior

Variational dropout (VD) (Kingma et al., 2015; Gal & Ghahramani, 2016; Molchanov et al., 2017; Hron et al., 2018; Liu et al., 2019) is a set of approaches that models the variational posterior distribution based on the dropout regularization technique. The dropout regularization randomly turns off some of the Neural Network (NN) parameters during training by multiplying discrete Bernoulli random noises to the parameters (Srivastava et al., 2014; Hinton et al., 2012). This technique was originally popularized as an efficient way to prevent NN model's over-fitting. Later, the fast dropout (Wang & Manning, 2013; Wan et al., 2013) reported that multiplying the continuous noises sampled from Gaussian distributions works similarly to the conventional Bernoulli dropout (Srivastava et al., 2014) due to the central limit theorem. The Gaussian approximation of the Bernoulli dropout is convenient since it provides a fully factorized (tractable) posterior model over the NN's parameters (Wang & Manning, 2013; Wan et al., 2013). In addition, VD approaches often utilize sparse priors for regularization (Kingma et al., 2015; Molchanov et al., 2017; Hron et al., 2018; 2017; Liu et al., 2019). This posterior and prior distribution modeling in VD enables the learning of each independent dropout rate over the NN parameters as variational parameters. This distinguishes the VD approaches from the conventional dropouts that use a single fixed rate over all parameters. However, the *dropout* posterior in the conventional VD becomes fixed once trained and it is unable to express the conditional posterior according to multiple tasks.

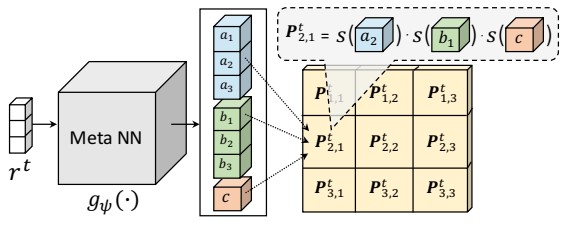 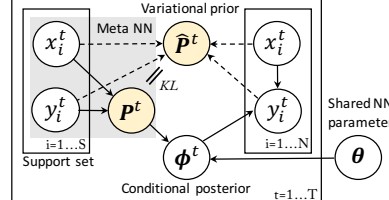

(a) The low-rank product of Bernoulli experts meta-model  (b) The probabilistic graphical model of NVDP's

Figure 1: (a) The low-rank product of Bernoulli experts meta-model of conditional *dropout* posterior. (b) The probabilistic graphical model of Neural Variational Dropout Processes (NVDPs) with the variational prior. Where $\{x_i^t, y_i^t\}_{i=1}^N$ is the N i.i.d samples from the $t$-th training dataset $\mathcal{D}^t$ among $T$ tasks. The context set $\mathcal{D}_C^t = \{x_i^t, y_i^t\}_{i=1}^S$ is a small subset of the $t$-th training dataset.

**Conditional dropout posterior.** We propose a new amortized variational posterior model that can be efficiently adapted for each given task. Suppose we train a fully connected NN of $L$ layers, then a conditional *dropout* posterior[3] based on the task-specific dropout rates $\mathbf{P}^t$ over the $K \times D$ dimensional deterministic parameters $\theta$ of each $l$-th layer of the NN can be given as follows:

$$q(\phi^t|\mathcal{D}_C^t; \theta) = \prod_{k=1}^K \prod_{d=1}^D q(\phi_{k,d}^t|\mathcal{D}_C^t) = \prod_{k=1}^K \prod_{d=1}^D \mathcal{N}(\phi_{k,d}^t|(1-\mathbf{P}_{k,d}^t)\theta_{k,d}, \mathbf{P}_{k,d}^t(1-\mathbf{P}_{k,d}^t)\theta_{k,d}^2). \quad (2)$$

The parameter[4] $\theta_{k,d}$ is shared across different tasks, representing the common task-invariant structure. In the equation 2, the task-specific NN parameter $\phi^t$ are fully described by the mean and variance of each independent Gaussian distribution via $\theta_{k,d}$ and $\mathbf{P}_{k,d}^t$. Note that the variational posterior model is explicitly conditioned on the subset of the $t$-th training dataset $\mathcal{D}_C^t = \{\mathbf{x}_i^t, \mathbf{y}_i^t\}_{i=1}^S$ ($\subseteq \mathcal{D}^t$) known as the $t$-th context set. The key idea of conditional posterior modeling in NVDPs is to employ an NN-based meta-model to predict the task-specific dropout rate $\mathbf{P}_{k,d}^t$ from the small context set $\mathcal{D}_C^t$. The meta-model to approximate $\mathbf{P}_{k,d}^t$ for each given task is simply defined as:

$$\mathbf{P}_{k,d}^t = s(\mathbf{a}_k) \cdot s(\mathbf{b}_d) \cdot s(\mathbf{c}), \text{ where } (\mathbf{a}, \mathbf{b}, \mathbf{c}) = g_\psi(r^t). \quad (3)$$

Here, the set representation $r^t$ is defined as the mean of features obtained from each data in $t$-th context set $\mathcal{D}_C^t$ (*i.e.*, $r^t = \sum_{i=1}^S h_\omega(\mathbf{x}_i^t, \mathbf{y}_i^t)/S$, where $h_\omega$ is a feature extracting NN parameterized by $\omega$), summarizing order invariant set information (Edwards & Storkey, 2017; Lee et al., 2019b). The $g_\psi(\cdot)$ is meta NN model parameterized by $\psi$ to predict a set of logit vectors (i.e., $\mathbf{a} \in \mathbb{R}^K, \mathbf{b} \in \mathbb{R}^D$ and $\mathbf{c} \in \mathbb{R}$). Then, the *sigmoid* function with a learnable temperature parameter $\tau$ (i.e., $s_\tau(\cdot) = 1/(1 + \exp(-(\cdot)/\tau))$ ) is applied to them to get the low-rank components of task-specific dropout rates $\mathbf{P}_{k,d}^t$: the row-wise $s(\mathbf{a}_k)$, column-wise $s(\mathbf{b}_d)$, and layer-wise dropout rate $s(\mathbf{c})$. In other words, the task-specific dropout rate $\mathbf{P}_{k,d}^t$ is obtained by multiplying low-rank components of the conditionally approximated dropout rates from the NN-based meta-model $g_\psi(\mathcal{D}_C^t)$ (see Figure1 (a)).

The product of $n$ Bernoulli random variables is also a Bernoulli variable (Leemis, 2020). By exploiting this property, we interpret the approximation of the task-specific dropout rates in terms of the low-rank product of Bernoulli experts. Unlike VERSA whose meta-model's complexity is $\mathcal{O}(LKD)$ to model the full NN weight posterior directly, the complexity of meta-model in NVDPs is $\mathcal{O}(L(K+D+1))$. In addition, the meta-model's role is only to predict the low-rank components of the task-specific dropout rates. With the shared parameter $\theta$, this can greatly reduce the complexity of the posterior distribution approximation of the high-dimensional task-specific NN's weights using only a few observed context examples. The product model tends to give sharp probability boundaries, which is often used for modeling the high dimensional data space (Hinton, 2002).

## 3.2 THE TASK-SPECIFIC VARIATIONAL PRIOR

To optimize the conditional *dropout* posterior in equation 2, a specification of the prior distribution $p(\phi^t)$ is necessary to get a tractable derivation of the KL regularization term of the ELBO in equa-

---

[3]The original Gaussian approximation in the VD is $q(\phi_{k,d}) = \mathcal{N}(\phi_{k,d}|\theta_{k,d}, \alpha\theta_{k,d}^2)$ with $\alpha = \mathbf{P}_{k,d}/(1-\mathbf{P}_{k,d})$. But, NVDPs extend the Bernoulli *dropout* model (Wang & Manning, 2013; Wan et al., 2013).

[4]We omit the layer index $l$ of the parameter $\theta_{l,k,d}$ for brevity.

tion 1. The question is how we can define the effective task-specific prior. In fact, an important requirement in the choice of the prior distribution in the conventional VD framework (Kingma et al., 2015) is that the analytical derivation of the KL term (*i.e.*, $\text{KL}(q(\phi^t|\mathcal{D}_C^t; \theta)||p(\phi^t))$ in equation 1) should not depend on the deterministic NN parameter $\theta$. This allows the constant optimization of the ELBO w.r.t all the independent variational parameters (*i.e.*, $\theta_{l,k,d}$ and $\mathbf{P}_{l,k,d}$ for all $l = 1 \ldots L$, $k = 1 \ldots K$ and $d = 1 \ldots D$). One way of modeling the prior is to employ the log-uniform prior $p(\log(|\phi|) \propto c$ as in the conventional VD (Kingma et al., 2015). However, a recently known limitation is that the log-uniform prior is an improper prior (*e.g.*, the KL divergence between the posterior and the log-uniform prior is infinite). This could yield a degeneration of the *dropout* posterior model to a deterministic one (Molchanov et al., 2017; Hron et al., 2018; 2017; Liu et al., 2019). Besides, the conventional prior used in VD approaches does not support a task-dependent regularization.

**Variational prior.** We introduce a new task-specific prior modeling approach to optimize the proposed conditional posterior; it is approximated with the *variational prior* defined by the same *dropout* posterior model in equation 2 except that the prior is conditioned on the whole task data (i.e., $p(\phi^t) \approx q(\phi^t|\mathcal{D}^t)$). The KL divergence between the conditional *dropout* posterior (with the only context set) and the *variational prior* (with the whole task data) can be derived as:

$$\text{KL}(q(\phi^t|\mathcal{D}_C^t)||q(\phi^t|\mathcal{D}^t)) = \sum_{k=1}^K \sum_{d=1}^D \{\frac{\mathbf{P}_{k,d}^t(1 - \mathbf{P}_{k,d}^t) + (\hat{\mathbf{P}}_{k,d}^t - \mathbf{P}_{k,d}^t)^2}{2\hat{\mathbf{P}}_{k,d}^t(1 - \hat{\mathbf{P}}_{k,d}^t)} + \frac{1}{2} \log \frac{\hat{\mathbf{P}}_{k,d}^t(1 - \hat{\mathbf{P}}_{k,d}^t)}{\mathbf{P}_{k,d}^t(1 - \mathbf{P}_{k,d}^t)}\}$$

$$(4)$$

where both $\mathbf{P}_{k,d}$ and $\hat{\mathbf{P}}_{k,d}$ are the dropout rates predicted from the meta-model but with different conditional set information: $\mathbf{P}_{k,d}$ is obtained from the small context set $\mathcal{D}_C^t$, while $\hat{\mathbf{P}}_{k,d}$ is from the whole task set $\mathcal{D}^t$. Interestingly, the analytical derivation of the KL is independent of the shared parameter $\theta$, thus this satisfies the necessary condition to be used as a prior in the VI optimization of the *dropout* posterior. Figure 1(b) depicts the variational prior is conditioned on the whole dataset.

The shared posterior model for the task-specific prior introduced here was inspired by recent Bayesian meta-learning approaches (Garnelo et al., 2018b; Kim et al., 2019; Iakovleva et al., 2020), but some critical differences are that: 1) we have developed the *variational prior* to regularize the task-specific dropout rates in the optimization of the conditional *dropout* posterior, 2) the denominator and the numerator of the KL divergence term in equation 4 are reversed compared with the existing approaches. In fact, some other recent studies of amortized VI inference (Tomczak & Welling, 2018; Takahashi et al., 2019) analytically derived that the optimal prior maximizing VI objective is the variational posterior aggregated on the whole dataset: $p^*(\phi) = \int_{\mathcal{D}} q(\phi|\mathcal{D})p(\mathcal{D})$. Thus, we hypothesized the conditional posterior model that depends on the whole dataset should be used to approximate the optimal prior $p^*(\phi^t) \approx q(\phi^t|\mathcal{D}^t)$ since the variational model conditioned on the aggregated set representation with much larger context (e.g., $\mathcal{D}^t \supseteq \mathcal{D}_C^t$) is likely to be much closer to the optimal task-specific prior than the model conditioned only on a subset. The experiments in Section 5 demonstrate that the proposed *variational prior* approach provides a reliable regularization for the conditional *dropout* posterior and the similar formulation is also applicable to the latent variable based conditional posterior models (Garnelo et al., 2018b; Kim et al., 2019).

### 3.3 STOCHASTIC VARIATIONAL INFERENCE

With the conditional dropout posterior defined in equation 2 and the KL regularization term in equation 4, we can now fully describe the ELBO objective of NVDPs for the multi-task dataset as:

$$\sum_{t=1}^T \log p(\mathcal{D}^t) \geq \sum_{t=1}^T \{\mathbb{E}_{q(\phi^t|\mathcal{D}_C^t; \theta)}[\log p(\mathbf{y}^t|\mathbf{x}^t, \phi^t)] - \text{KL}(q(\phi^t|\mathcal{D}_C^t; \theta)||q(\phi^t|\mathcal{D}^t; \theta))\}. \quad (5)$$

The goal is to maximize the ELBO equation 5 w.r.t. the variational parameters (*i.e.* $\theta$, $\psi$, $\omega$, and $\tau$) of the posterior $q(\phi_t|\mathcal{D}_C^t; \theta)$ defined in equation 2. The optimization of these parameters is done by using the *stochastic gradient variational Bayes* (SGVB) (Kingma & Welling, 2013; Kingma et al., 2014). The basic trick in the SGVB is to parameterize the random weights $\phi^t \sim q(\phi^t|\mathcal{D}^t)$ using a deterministic differentiable transformation $\phi^t = f(\epsilon, \mathcal{D}_C^t)$ with a non-parametric i.i.d. noise $\epsilon \sim p(\epsilon)$. Then, an unbiased differentiable minibatch-based Monte Carlo estimator, $\hat{\mathcal{L}}(\theta, \psi, \omega, \tau)$,

of the ELBO of NVDPs can be defined as:

$$\frac{1}{T'}\sum_{t=1}^{T'}\{\frac{1}{M}\sum_{i=1}^{M}\log p(\mathbf{y}_i^t|\mathbf{x}_i^t, f(\epsilon, \mathcal{D}_C^t)) - (\frac{\mathbf{P}^t(1-\mathbf{P}^t)+(\hat{\mathbf{P}}^t-\mathbf{P}^t)^2}{2\hat{\mathbf{P}}^t(1-\hat{\mathbf{P}}^t)} + \frac{1}{2}\log\frac{\hat{\mathbf{P}}^t(1-\hat{\mathbf{P}}^t)}{\mathbf{P}^t(1-\mathbf{P}^t)})\} \quad (6)$$

$T'$ and $M$ are the size of randomly sampled mini-batch of tasks and data points, respectively, per each epoch (*i.e.*, $\{\mathbf{x}_i^t, \mathbf{y}_i^t\}_{i=1}^M = \mathcal{D}^t \sim \{\mathcal{D}^t\}_{t=1}^{T'} \sim \mathcal{D})$. $\mathcal{D}_C^t$ is the subset set of $\mathcal{D}^t$ discussed in section 3.1 and 3.2. The transformation of the task-specific weights is given as $\phi_{k,d}^t = f(\epsilon_{k,d}, \mathcal{D}_C^t) = (1-\mathbf{P}_{k,d}^t)\theta_{k,d} + \sqrt{\mathbf{P}_{k,d}^t(1-\mathbf{P}_{k,d}^t)}\theta_{k,d}\epsilon_{k,d}$ with $\epsilon_{k,d} \sim N(0,1)$ from the equation 2. The intermediate weight $\phi_{k,d}^t$ is now differentiable with respect to $\theta_{k,d}$ and $\mathbf{P}_{k,d}^t$. Also, $\mathbf{P}_{k,d}^t$ (and $\hat{\mathbf{P}}_{k,d}^t$) is deterministically computed by the meta-model parameterized by $\psi, \omega$, and $\tau$ as in equation 3. Thus, the estimator $\hat{\mathcal{L}}(\theta, \psi, \omega, \tau)$ in equation 6 is differentiable with respect to all the variational parameters and can be optimized via the SGD algorithm. In practice, a local reparameterization trick[5] is further utilized to reduce the variance of gradient estimators on training (Kingma et al., 2015).

## 4 RELATED WORKS

Gaussian Processes(GPs) (Rasmussen & Williams, 2005) are classical Bayesian learning approaches closely related to NVDPs; GPs allow an analytical derivation of the posterior predictive distribution given observed context points based on Gaussian priors and kernels. However, classical GPs require an additional optimization procedure to identify the suitable kernel for each task, and the time complexity is quadratic to the number of contexts. Recently introduced model-based Bayesian meta-learning methods (Garnelo et al., 2018b;a; Kim et al., 2019; Gordon et al., 2018; Iakovleva et al., 2020) offer NN-based conditional posterior models that can be efficiently learned from data and later provide an instant posterior predictive estimation at test time. One important aspect of these model-based methods have in common is the utilization of *permutation invariant* set representation as an input to the meta-model (Edwards & Storkey, 2017; Lee et al., 2019b; Bloem-Reddy & Teh, 2020; Teh & Lecture, 2019), which makes a conditional posterior invariant to the order of the observed context data. This property, known as *exchangeability*, is a necessary condition to define a stochastic process according to Kolmogorov's Extension Theorem (Garnelo et al., 2018b; Kim et al., 2019; Øksendal, 2003). NVDPs also utilize the set representation to approximate the task-specific dropout rates in the conditional *dropout* posterior. Another related work to NVDPs is Meta Dropout (Lee et al., 2019a). They also proposed a unique input-dependent dropout approach; Gaussian-based stochastic perturbation is applied to each preactivation feature of NN functions. Their approach, however, is constructed using optimization-based meta-learning methods (Finn et al., 2017; Li et al., 2017) without directly considering the set representation. NVDPs, employing set representation, extend the *dropout* posterior of Variational Dropout (VD) (Kingma et al., 2015; Wang & Manning, 2013; Wan et al., 2013; Liu et al., 2019; Molchanov et al., 2017) in the context of model-based meta-learning and introduce a new idea of *variational prior* that can be universally applied to other Bayesian learning approaches (Garnelo et al., 2018b; Kim et al., 2019). Gal & Ghahramani (2016) also discussed the theoretical connection between VD and GPs.

## 5 EXPERIMENTS

**Metrics in regression.** In the evaluation of the conditional NN models' adaptation, the newly observed task data $\mathcal{D}^*$ is split into the input-output pairs of the *context set* $\mathcal{D}_C^* = \{x_i, y_i\}_{i=1}^S$ and the *target set* $\mathcal{D}_T^* = \{x_i, y_i\}_{i=1}^N$ ($\mathcal{D}_C^* \not\subseteq \mathcal{D}_T^*$ in evaluation (Garnelo et al., 2018b; Gordon et al., 2018)). (1) The log-likelihood (LL), $\frac{1}{N+S}\sum_{i\in\mathcal{D}_C^*\cup\mathcal{D}_T^*}\mathbb{E}_{q(\phi^*|\mathcal{D}_C^*)}[\log p(y_i|x_i,\phi^*)]$, measures the performance of the NN model over the whole dataset $\mathcal{D}^*$ conditioned on the *context set*. (2) The reconstructive log-likelihood (RLL), $\frac{1}{S}\sum_{i\in\mathcal{D}_C^*}\mathbb{E}_{q(\phi^*|\mathcal{D}_C^*)}[\log p(y_i|x_i,\phi^*)]$, measures how well the model reconstructs the data points in the *context set*. A low RLL is a sign of under-fitting. (3) The predictive log-likelihood (PLL), $\frac{1}{N}\sum_{i\in\mathcal{D}_T^*}\mathbb{E}_{q(\phi^*|\mathcal{D}_C^*)}[\log p(y_i|x_i,\phi^*)]$, measures the prediction on the data points in the *target set* (not in the *context set*). A low PLL is a sign of over-fitting.

---

[5]We can sample pre-activations $B_{m,d}$ for a mini-batch of size $M$ directly using inputs $A_{m,k}$ ($B_{m,d} \sim \mathcal{N}(B_{m,d}|\sum_{k=1}^K A_{m,k}(1-\mathbf{P}_{k,d}^t)\theta_{k,d}, \sum_{k=1}^K A_{m,k}^2 \mathbf{P}_{k,d}^t(1-\mathbf{P}_{k,d}^t)\theta_{k,d}^2)$).

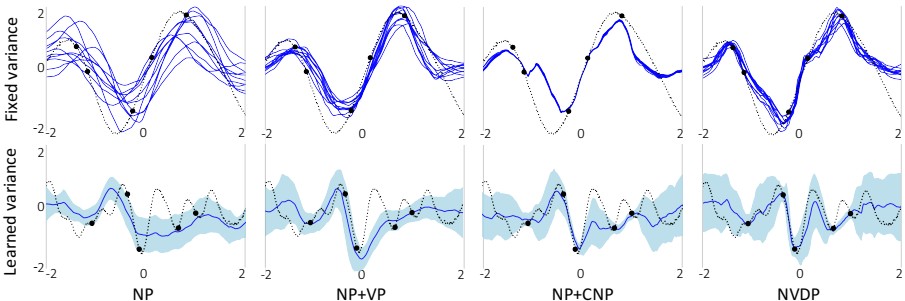

Figure 2: The 1D few-shot regression results of the models on GP dataset in *fixed variance* and *learned variance* settings. The black (dash-line) represent the true unknown task function. Black dots are a few context points ($S = 5$) given to the posteriors. The blue lines (and light blue area in *learned variance* settings) are mean values (and variance) predicted from the sampled NNs.

| GP Dataset | | NP | NP+VP | NP+CNP | NP+CNP+VP | NVDP |
|---|---|---|---|---|---|---|
| *Fixed* | LL | $-0.98(\pm 0.08)$ | $-0.96(\pm 0.06)$ | $-0.95(\pm 0.05)$ | $-0.94(\pm 0.05)$ | $\mathbf{-0.94(\pm 0.04)}$ |
| *Variance* | RLL | $-0.97(\pm 0.06)$ | $-0.95(\pm 0.04)$ | $-0.93(\pm 0.02)$ | $-0.93(\pm 0.02)$ | $\mathbf{-0.93(\pm 0.01)}$ |
| | PLL | $-0.98(\pm 0.08)$ | $-0.97(\pm 0.06)$ | $-0.95(\pm 0.05)$ | $-0.94(\pm 0.05)$ | $\mathbf{-0.94(\pm 0.05)}$ |
| *Learned* | LL | $0.19(\pm 1.87)$ | $0.48(\pm 0.77)$ | $0.70(\pm 0.89)$ | $0.70(\pm 0.73)$ | $\mathbf{0.83(\pm 0.61)}$ |
| *Variance* | RLL | $0.72(\pm 0.57)$ | $0.71(\pm 0.55)$ | $1.03(\pm 0.39)$ | $1.00(\pm 0.41)$ | $\mathbf{1.10(\pm 0.34)}$ |
| | PLL | $0.16(\pm 1.90)$ | $0.46(\pm 0.78)$ | $0.66(\pm 0.92)$ | $0.68(\pm 0.75)$ | $\mathbf{0.81(\pm 0.62)}$ |

Table 1: The validation result of the 1D regression models on the GP dataset in *fixed variance* and *learned variance* settings. The higher LLs are the better. The models of NP, NP with variational prior (NP+VP), NP with deterministic path (NP+CNP), NP+CNP+VP and NVDP (ours) are compared.

**GP Samples.** The 1D regression task is to predict unknown functions given some observed context points; each function (or data point) is generated from a Gaussian Process (GP) with random kernel, then the standard data split procedure is performed (i.e. $S \sim U(3, 97)$ and $N \sim U[S + 1, 100))$ at train time. For the baseline models, Neural Process model (NP) and NP with additional deterministic representation (NP+CNP) described in (Garnelo et al., 2018b;a; Kim et al., 2019) is compared. We also adopted the variational prior (VP) into the representation based posterior of NP and its variants (NP+VP and NP+CNP+VP). For all models (including our NVDP), we used the same *fixed variance* and *learned variance* likelihood architecture depicted in (Kim et al., 2019): the agent NN with 4 hidden layers of 128 units with LeLU activation (Nair & Hinton, 2010) and an output layer of 1 unit for mean (or an additional 1 unit for variance). The dimensions of the set representation $r^t$ were fixed to 128. The meta NNs in the conditional *dropout* posterior in NVDP has 4 hidden layers of 128 units with LeakyReLU and an output layer of 257 units (i.e. $K+D+1$) for each layer of the agent model. All models were trained with Adam optimizer (Kingma & Ba, 2015) with learning rate 5e-4 and 16 task-batches for 0.5 million iterations. On validation, 50000 random tasks (or functions) were sampled from the GP function generator, and the split data of $S \sim U(3, 97)$ and $N = 400 - S$ were used to compute the log-likelihood (LL) and other evaluation metrics.

Table 1 summarizes the validation results of 1D regression with the GP dataset. NVDP achieves the best LL scores compared with all other baselines in both the *fixed* and *learned variance* likelihood model settings. The NVDPs record high RLL on the observed data points and excellent PLL scores on the unseen function space in the new task; this indicates that the proposed conditional *dropout* posterior approach can simultaneously mitigate the under-fitting and over-fitting of the agent model compared with the other baselines. When VP is applied to NP or NP+CNP, the PLL scores tend to increase by meaningful margins in all cases. This demonstrates that the proposed variational prior (VP) approach can also reduce the over-fitting of latent representation-based conditional posterior. Figure 2 visualizes the few-shot 1D function regression results in both model settings. In the *fixed variance* setting, the functions sampled from NPs show high variability but cannot fit the context data well. NP+CNP can fit the context data well but loses *epistemic* uncertainty due to the collapsing of conditional posterior. On the other hand, the functions from NVDPs show a different behavior; they capture the function's variability well while also fitting the observed context points better. NVDPs also approximate the mean and variance of unknown function well in *learned variance* setting.

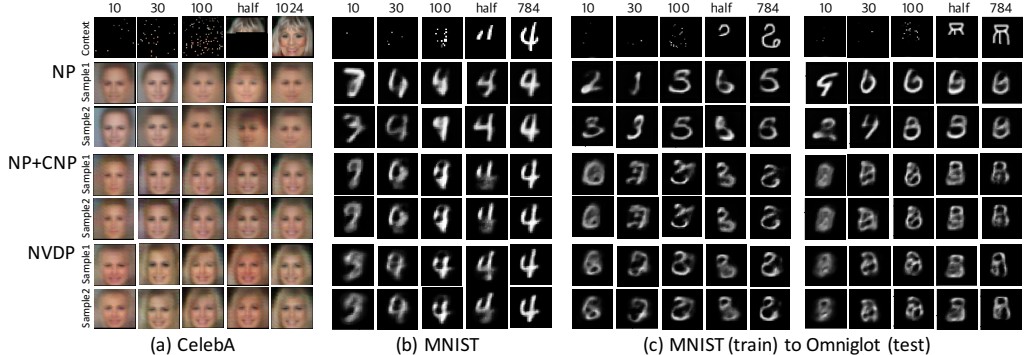

Figure 4: The results from the 2D image completion tasks on CelebA, MNIST, and Omniglot dataset. Given the observed context points (10, 30, 100, half, and full pixels), the mean values of two independently sampled functions from the models (i.e. NP, NP+CNP and NVDP (ours)) are presented.

| Image Dataset | | NP | NP+VP | NP+CNP | NP+CNP+VP | NVDP |
|---|---|---|---|---|---|---|
| *MNIST* | LL | 0.54($\pm$0.51) | 0.76($\pm$0.14) | 0.83($\pm$0.21) | 0.86($\pm$0.16) | **0.90($\pm$0.16)** |
| | RLL | 0.94($\pm$0.18) | 0.90($\pm$0.10) | 1.12($\pm$0.09) | 1.12($\pm$0.09) | **1.15($\pm$0.05)** |
| | PLL | 0.51($\pm$0.51) | 0.75($\pm$0.14) | 0.80($\pm$0.20) | 0.83($\pm$0.15) | **0.88($\pm$0.15)** |
| *MNIST (train)* | LL | 0.35($\pm$0.29) | 0.56($\pm$0.10) | 0.64($\pm$0.17) | 0.68($\pm$0.13) | **0.70($\pm$0.13)** |
| to | RLL | 0.72($\pm$0.19) | 0.73($\pm$0.12) | 0.95($\pm$0.09) | 0.98($\pm$0.09) | **0.99($\pm$0.07)** |
| *Omniglot (test)* | PLL | 0.32($\pm$0.28) | 0.54($\pm$0.11) | 0.60($\pm$0.16) | 0.64($\pm$0.13) | **0.66($\pm$0.12)** |
| *CelebA* | LL | 0.51($\pm$0.27) | 0.60($\pm$0.12) | 0.76($\pm$0.15) | 0.77($\pm$0.15) | **0.83($\pm$0.15)** |
| | RLL | 0.63($\pm$0.11) | 0.68($\pm$0.06) | 0.91($\pm$0.05) | 0.92($\pm$0.05) | **0.99($\pm$0.04)** |
| | PLL | 0.50($\pm$0.27) | 0.59($\pm$0.12) | 0.75($\pm$0.15) | 0.76($\pm$0.15) | **0.82($\pm$0.15)** |

Table 2: The summary of 2D image completion tasks on the MNIST, CelebA, and Omniglot dataset.

**Active learning with regression.** To further compare the uncertainty modeling accuracy, we performed an additional active learning experiment on the GP dataset described above. The goal in active learning is to improve the log-likelihood of models with a minimal number of context points. To this end, each model chooses additional data points sequentially; the points with maximal variance across the sampled regressors were selected at each step in our experiment. The initial data point is randomly sampled within the input domain, followed by 19 additional points that are selected according to the variance estimates. As seen in Figure 3, NVDPs outperform the others due to their accurate variance estimation, especially with a small number of additional points, and show steady improvement with less over-fitting.

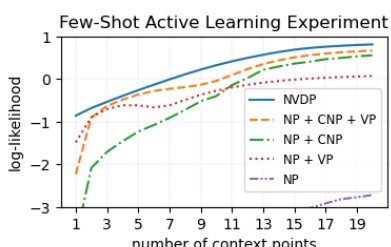

Figure 3: Active learning performance on regression after up to 19 selected data points. NVDPs can use its uncertainty estimation to quickly improve LLs, while other models are learning slowly.

**Image completion tasks.** The image completion tasks are performed to validate the performance of the models in the more complex function spaces (Garnelo et al., 2018b;a; Kim et al., 2019). Here, we treat the image samples from MNIST (LeCun et al., 1998) and CelebA (Liu et al., 2015) as unknown functions. The task is to predict a mapping from normalized 2D pixel coordinates $x_i$ ($\in [0,1]^2$) to pixel intensities $y_i$ ($\in \mathbb{R}^1$ for greyscale, $\in \mathbb{R}^3$ for RGB) given some context points. We used the same *learned variance* baselines implemented in the GP data regression task (except the 2D input and 3D output for the agent NN and $r^t = 1024$ were used for CelebA). At each iteration, the images in the training set are split into $S$ context and $N$ target points (e.g., $S \sim U(3, 197)$, $n \sim U[N+1, 200)$ at train and $S \sim U(3, 197)$, $N = 784 - S$ at validation). Adam optimizer with a learning rate 4e-4 and 16 task batches with 300 epochs were used for training. The validation is performed on the separated validation set. To see the generalization performance on a completely new dataset, we also tested the models trained on MNIST to the Omniglot validation set.

Table 2 summarizes the validation results of image completion tasks. NVDPs achieve the outperforming LLs compared with all other baselines. Interestingly, NVDPs (trained on the MNIST

| Method | *Omniglot* dataset | | | | *MiniImageNet* dataset | |
| | 5-way accuracy (%) | | 20-way accuracy (%) | | 5-way accuracy (%) | |
| | 1-shot | 5-shot | 1-shot | 5-shot | 1-shot | 5-shot |
| --- | --- | --- | --- | --- | --- | --- |
| Matching Nets | 98.1 | 98.9 | 93.8 | 98.5 | 46.6 | 60.0 |
| Prototypical Nets | 97.4 | 99.3 | 95.4 | 98.7 | $46.61 \pm 0.78$ | $65.77 \pm 0.70$ |
| CNP | 95.3 | 98.5 | 89.9 | 96.8 | $48.05 \pm 2.85$ | $62.71 \pm 0.58$ |
| Meta-SGD (MSGD) | - | - | $96.16 \pm 0.14$ | $98.54 \pm 0.07$ | $48.30 \pm 0.64$ | $65.55 \pm 0.56$ |
| MSGD + Meta-dropout | - | - | $97.02 \pm 0.13$ | $\mathbf{99.05 \pm 0.05}$ | $50.87 \pm 0.63$ | $65.55 \pm 0.57$ |
| MAML | $98.70 \pm 0.40$ | $\mathbf{99.90 \pm 0.10}$ | $95.80 \pm 0.63$ | $98.90 \pm 0.20$ | $48.70 \pm 1.84$ | $63.11 \pm 0.92$ |
| VERSA | $99.70 \pm 0.20$ | $99.75 \pm 0.13$ | $97.66 \pm 0.29$ | $98.77 \pm 0.18$ | $53.40 \pm 1.82$ | $67.37 \pm 0.86$ |
| NVDP | $\mathbf{99.70 \pm 0.12}$ | $99.86 \pm 0.28$ | $\mathbf{97.98 \pm 0.22}$ | $98.99 \pm 0.22$ | $\mathbf{54.06 \pm 1.86}$ | $\mathbf{68.12 \pm 1.04}$ |

Table 3: Few-shot classification results on Omniglot and MiniImageNet dataset. The baselines are Matching Nets, Prototypical Nets, MAML, Meta-SGD, Meta-dropout, CNP, VERSA, and NVDP (our). Each value corresponds to the classification accuracy (%) (and *std*) on validation set.

dataset) also achieve the best results on the Omniglot dataset. Figure 4 shows the image reconstruction results with a varying number of random context pixels. NP generated various image samples when the number of contexts was small (e.g., $m \leq 30$ or half), but those samples could not approximate the true unknown images well compared with the other models. The NP+CNP achieves crisp reconstruction results compared with NP and also shows a good generalization performance on the Omniglot dataset, but the sampled images (or functions) from NP+CNP had almost no variability due to its posterior collapsing behavior. On the other hand, the samples from NVDPs exhibit comparable reconstruction results while also showing a reasonable amount of variability. In addition, NVDPs also present an outstanding generalization performance on the unseen Omniglot dataset. This implies that NVDPs not only fit better in the complex function regression but can also capture more general knowledge that can be applied to new unseen tasks.

**Few-shot classification tasks.** NVDPs can also be successfully applied for few-shot classification tasks; we have tested NVDPs on standard benchmark datasets such as Omniglot (Lake et al., 2015) and MiniImagenet (Ravi & Larochelle, 2017) with other baselines: VERSA (Gordon et al., 2018), CNP (Garnelo et al., 2018a), Matching Nets (Vinyals et al., 2016), Prototypical Nets (Snell et al., 2017), MAML (Finn et al., 2017), Meta-SGD (Li et al., 2017) and Meta-dropout (Lee et al., 2019a). For the classifier, we used one-layer NNs with hidden units of 512. For the meta-model, we used two-layer NNs with hidden units of 256 similar to VERSA's conditional posterior model. For an image input, the NN classifier outputs one logit value per each class; class-specific dropout rates for the NN classifier are computed with the image features $r^t \in \mathbb{R}^{256}$ aggregated by the same class in the few-shot context examples. The same deep (CONV5) feature extractor architecture is used as in VERSA (Gordon et al., 2018). For each class, 1 or 5 few-shot context samples (i.e., labeled images) are given. Among 5 or 20 classes, only the logit value related to the true label is maximized. We use the same batch sizes, learning rate, and epoch settings depicted in VERSA (Gordon et al., 2018). Table 3 summarizes the results. NVDPs achieve higher predictive accuracy than the model-based meta-learning approaches CNP and VERSA and is comparable with the state-of-the-art optimization-based meta-learning approaches such as MAML or Meta-Dropout on the Omniglot dataset. NVDPs also record a good classification accuracy in the MiniImageNet dataset.

## 6 CONCLUSION

This study presents a new model-based Bayesian meta-learning approach, Neural Variational Dropout Processes (NVDPs). A novel conditional *dropout* posterior is induced from a meta-model that predicts the task-specific dropout rates of each NN parameter conditioned on the observed context. This paper also introduces a new type of *variational prior* for optimizing the conditional posterior in the amortized variational inference. We have evaluated the proposed method compared with the existing approaches in various few-shot learning tasks, including 1D regression, image inpainting, and classification tasks. The experimental results demonstrate that NVDPs simultaneously improved the model's adaptation to the context data, functional variability, and generalization to new tasks. The proposed *variational prior* could also improve the variability of the representation-based posterior model (Garnelo et al., 2018b). Adapting the advanced set representations of (Lee et al., 2019b; Kim et al., 2019; Volpp et al., 2020) or investigating more complex architectures (Chen et al., 2019; Gordon et al., 2020; Foong et al., 2020) for NVDPs would be interesting future work.

ACKNOWLEDGMENT

This work was supported by Center for Applied Research in Artificial Intelligence(CARAI) grant funded by Defense Acquisition Program Administration(DAPA) and Agency for Defense Development(ADD) (UD190031RD) and Basic Science Research Program through National Research Foundation of Korea (NRF) funded by the Korea government (MSIT) (NRF-2020R1A2B5B03095585). Gunhee Kim is the corresponding author. Insu Jeon thanks Jaesik Yoon, Minui Hong, and Kangmo Kim for helpful comments.

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

## A  DERIVATION OF THE ELBO OF NVDPs

This section describes a detailed derivation of the evidence lower-bound (ELBO) of NVDPs in equation 5. Given the context data $\mathcal{D}_C^t = (\mathbf{x}_C^t, \mathbf{y}_C^t)$ and target data $\mathcal{D}^t = (\mathbf{x}^t, \mathbf{y}^t)$, the KL divergence between the true unknown posterior distribution over parameter $p(\phi^t|\mathbf{x}^t, \mathbf{y}^t)$ and the (conditional) variational posterior $q(\phi^t|\mathcal{D}_C^t)$ is given by:

$$
\sum_{t=1}^{T} D_{\text{KL}}(q(\phi^t|\mathcal{D}_C^t)||p(\phi^t|\mathcal{D}^t)) = \sum_{t=1}^{T} \int q(\phi^t|\mathcal{D}_C^t) \log \frac{q(\phi^t|\mathcal{D}_C^t)}{p(\phi^t|\mathbf{x}^t, \mathbf{y}^t)} d\phi^t
$$

$$
= \sum_{t=1}^{T} \int q(\phi^t|\mathcal{D}_C^t) \log \frac{q(\phi^t|\mathcal{D}_C^t)p(\mathbf{y}^t|\mathbf{x}^t)}{p(\mathbf{y}^t|\mathbf{x}^t, \phi^t)p(\phi^t)} d\phi^t \tag{7}
$$

$$
= \sum_{t=1}^{T} \int q(\phi^t|\mathcal{D}_C^t) \Big\{ \log \frac{q(\phi^t|\mathcal{D}_C^t)}{p(\phi^t)} + \log p(\mathbf{y}^t|\mathbf{x}^t) - \log p(\mathbf{y}^t|\phi^t, \mathbf{x}^t) \Big\} d\phi^t
$$

$$
= \sum_{t=1}^{T} D_{\text{KL}}(q(\phi^t|\mathcal{D}_C^t)||p(\phi^t)) + \log p(\mathbf{y}^t|\mathbf{x}^t) - \mathbb{E}_{q(\phi^t|\mathcal{D}_C^t)}[\log p(\mathbf{y}^t|\phi^t, \mathbf{x}^t)]. \tag{8}
$$

The step (7) is due to Bayes rule of $p(\phi^t|\mathbf{x}^t, \mathbf{y}^t) = \frac{p(\mathbf{y}^t|\phi^t, \mathbf{x}^t)p(\phi^t)}{p(\mathbf{y}^t|\mathbf{x}^t)}$ where $\phi^t$ is often assumed to be independent of $\mathbf{x}^t$. By reordering (8), we get

$$
\sum_{t=1}^{T} \log p(\mathbf{y}^t|\mathbf{x}^t) \geq \sum_{t=1}^{T} \mathbb{E}_{q(\phi^t|\mathcal{D}_C^t)}[\log p(\mathbf{y}^t|\mathbf{x}^t, \phi^t)] - D_{\text{KL}}(q(\phi^t|\mathcal{D}_C^t)||p(\phi^t)) \tag{9}
$$

$$
\approx \sum_{t=1}^{T} \mathbb{E}_{q(\phi^t|\mathcal{D}_C^t)}[\log p(\mathbf{y}^t|\phi^t, \mathbf{x}^t, \mathcal{D}_C^t)] - D_{\text{KL}}(q(\phi^t|\mathcal{D}_C^t)||q(\phi^t|\mathcal{D}^t)) \tag{10}
$$

The lower-bound in (9) is due to the non-negativity of the omitted term $\sum_{t=1}^{T} D_{\text{KL}}(q(\phi^t|\mathcal{D}_C^t)||p(\phi^t|\mathcal{D}^t))$ In the lower-bound of (9), the choice of $p(\phi^t)$ is often difficult since the biased prior can lead to over-fitting or under-fitting of the model. However, some recent studies of the amortized VI inference (Tomczak & Welling, 2018; Takahashi et al., 2019) analytically discussed that the optimal prior in the amortized variational inference is the aggregated conditional posterior model on whole dataset: $p^*(\phi) = \int_{\mathcal{D}} q(\phi|\mathcal{D})p(\mathcal{D})$. Usually, the aggregated posterior cannot be calculated in a closed form due to the expensive computation cost of the integral. However, the aggregation on the whole dataset in the model-based conditional posterior is constructed based on the set representation. This motivates us to define the conditional posterior given the whole task dataset as an empirical approximation of the optimal prior (i.e., $p^*(\phi^t) \approx q(\phi^t|\mathcal{D}^t)$). We call this a *variational prior*. Thus, the approximation of $p^t(\phi)$ in (9) with the *variational prior* yields the approximated lower bound of (10).

## B  DERIVATION OF THE KL DIVERGENCE IN THE ELBO OF NVDPs

An essential requirement in the choice of the prior is that the analytical derivation of the KL divergence term in equation (10) should not depend on the deterministic NN parameter $\theta$ (Kingma et al., 2015; Molchanov et al., 2017; Hron et al., 2018; 2017; Liu et al., 2019). This allows the constant optimization of the ELBO w.r.t all the independent variational parameters (*i.e.* $\theta_{l,k,d}$ and $\mathbf{P}_{l,k,d}$ for

all $l = 1 \ldots L$, $k = 1 \ldots K$, and $d = 1 \ldots D$). In fact, the conditional posterior defined in (2) is a Gaussian distribution, thus $\mathrm{KL}(q(\phi^t|\mathcal{D}_C^t;\theta)||q(\phi^t|\mathcal{D}^t;\theta))$ is analytically defined as:

$$D_{\mathrm{KL}}(q(\phi^t|\mathcal{D}_C^t;\theta)||q(\phi^t|\mathcal{D}^t;\theta)))$$

$$= \frac{\mathbf{P}^t(1-\mathbf{P}^t)\theta 2 + ((1-\mathbf{P}^t)\theta - (1-\hat{\mathbf{P}}^t)\theta)^2}{2\hat{\mathbf{P}}^t(1-\hat{\mathbf{P}}^t)\theta 2} + \log \frac{\sqrt{\hat{\mathbf{P}}^t(1-\hat{\mathbf{P}}^t)\theta 2}}{\sqrt{\mathbf{P}^t(1-\mathbf{P}^t)\theta 2}} - \frac{1}{2} \tag{11}$$

$$= \frac{\mathbf{P}^t(1-\mathbf{P}^t)\cancel{\theta 2} + (\hat{\mathbf{P}}^t - \mathbf{P}^t)^2\cancel{\theta 2}}{2\hat{\mathbf{P}}^t(1-\hat{\mathbf{P}}^t)\cancel{\theta 2}} + \frac{1}{2}\log \frac{\hat{\mathbf{P}}^t(1-\hat{\mathbf{P}}^t)\cancel{\theta 2}}{\mathbf{P}^t(1-\mathbf{P}^t)\cancel{\theta 2}} - \frac{1}{2} \tag{12}$$

$$= \frac{\mathbf{P}^t(1-\mathbf{P}^t) + (\hat{\mathbf{P}}^t - \mathbf{P}^t)^2}{2\hat{\mathbf{P}}^t(1-\hat{\mathbf{P}}^t)} + \frac{1}{2}\log \frac{\hat{\mathbf{P}}^t(1-\hat{\mathbf{P}}^t)}{\mathbf{P}^t(1-\mathbf{P}^t)} - \frac{1}{2} \tag{13}$$

where both $\mathbf{P}$ and $\hat{\mathbf{P}}$ are the dropout rates predicted from the meta-model via the equation (3) but with different conditional set information: $\mathbf{P}$ is obtained from the small context set $\mathcal{D}_C^t$, while $\hat{\mathbf{P}}$ is from the whole task set $\mathcal{D}^t$. (11) is derived using the analytical formulation of KL divergence between two Gaussian distributions. (13) is equivalent to the KL term defined in (4) of the manuscript (except that the constant term 1/2 is omitted for brevity). Interestingly, the analytical derivation of the KL is independent of the shared parameter $\theta$, thus this satisfies the necessary condition to be used as a prior in the VI optimization of the *dropout* posterior.

The KL divergence in (13) intuitively means that the dropout rates predicted from a small context set should be close to the dropout rates predicted from a much larger context set while training. The experiments validated that this surprisingly works well to induce a robust conditional functional uncertainty. However, one practical issue while training the dropout rate with the KL term (13) is that the dropout rate could converge to zeros during the early training period due to the larger gradients from the KL than from the likelihood [6]. In practice, we adopt the dropout rate clipping technique often used in other Variational Dropout approaches (Kingma et al., 2015; Hron et al., 2018; Liu et al., 2019; Molchanov et al., 2017; Gal & Ghahramani, 2016; Liu et al., 2019). We use the dropout rate range of (0.01, 0.99) for all experiments.

## C  ADDITIONAL EXPERIMENTAL DETAILS ON THE 1D REGRESSION.

**Setup.**   We explored the 1D function regression on the data generated from the synthetic GPs with varying kernels[7] in the previous work (Garnelo et al., 2018b;a; Kim et al., 2019) that is suitable for measuring the uncertainty adaptation. For the baseline models, Neural Process model (NP) and NP with additional deterministic representation (NP+CNP) described in (Garnelo et al., 2018b;a; Kim et al., 2019) is compared. We also adopted the variational prior (VP) into the representation-based posterior of NP and its variants (NP+VP and NP+CNP+VP). For all models (including our NVDP), we used the same *fixed variance* and *learned variance* likelihood architecture depicted in (Kim et al., 2019): the agent NN with 4 hidden layers of 128 units with LeLU activation (Nair & Hinton, 2010) and an output layer of 1 unit for mean (or an additional 1 unit for variance). The dimensions of the set representation $r^t$ were fixed to 128. The meta NNs in the conditional *dropout* posterior in NVDPs have 4 hidden layers of 128 units with LeakyReLU and an output layer of 257 units (i.e. $K+D+1$) for each layer of the agent model. All models were trained with Adam optimizer (Kingma & Ba, 2015) with learning rate 5e-4 for 0.5 million iterations. We draw 16 functions (a batch) from GPs at each iteration. Specifically, at every training step, we draw 16 functions $f(\cdot)$ from GP prior with the squared-exponential kernels, $k(x,x') = \sigma_f^2 \exp(-(x-x')^2/2l^2)$, generated with length-scale $l \sim U(0.1, 0.6)$ and function noise level $\sigma_f \sim U(0.1, 1.0)$. Then, $x$ values was uniformly sampled from $[-2, 2]$, and corresponding $y$ value was determined by the randomly drawn function (i.e. $y = f(x), f \sim \mathcal{GP}$). And the task data points are split into a disjoint sample of $m$ contexts and $n$ targets as $m \sim U(3, 97)$ and $n \sim U[m+1, 100]$, respectively. In the test or validation, the numbers of contexts and targets were chosen as $m \sim U(3, 97)$ and $n = 400 - m$, respectively. 50000 functions were sampled from GPs to compute the log-likelihood (LL) and other scores for validation.

---

[6]It also turns out to be a floating-point exception problem. We later set the minimum noise (i.e., *eps*) to $1e\text{-}10$ in the calculation of log, sqrt, and division function, and the collapsing problem did not occur again.

[7]`https://github.com/deepmind/neural-processes`

The architecture of NVDP model is as follows:

**(Deterministic) Feature Encoder.** $r(\mathbf{x}_s^t, \mathbf{y}_s^t) : 2 \times |C| \underbrace{\overset{\text{lin+relu}}{\longrightarrow} 128 \times |\mathcal{C}|}_{\text{6 times}} \overset{\text{mean}}{\longrightarrow} 128.$

**Decoder (Agent).** $f(x_i) : 1 \underbrace{\overset{\text{lin+relu}}{\longrightarrow} 128}_{\text{4 times}} \overset{\text{lin+relu}}{\longrightarrow} 2 \overset{\text{split}}{\longrightarrow} (\mu, \sigma)$ (where $\sigma = 0.1 + 0.9 \cdot$ softplus(logstd)).

**Meta Model.** $g^{(l)}(r) : d_r \underbrace{\overset{\text{lin+leakyReLu}}{\longrightarrow} 128}_{\text{4 times}} \overset{\text{lin+leakyReLu}}{\longrightarrow} (K^{(l)}, D^{(l)}, 1) \overset{\text{split}}{\longrightarrow} (\mathbf{a}, \mathbf{b}, \mathbf{c})$

where $g^{(l)}(r)$ is the meta NN for the $l$-th layer of the decoder, and $K^{(l)} \times D^{(l)}$ is the number of parameters in the $l$-th layer.

| GP Dataset | | CNP |
|---|---|---|
| *Fixed* | LL | $-0.94(\pm 0.05)$ |
| *Variance* | RLL | $-0.93(\pm 0.01)$ |
| | PLL | $-0.94(\pm 0.05)$ |
| *Learned* | LL | $0.72(\pm 0.54)$ |
| *Variance* | RLL | $1.03(\pm 0.38)$ |
| | PLL | $0.69(\pm 0.53)$ |

Table 4: An additional summary of the 1D regression with the GP with random kernel dataset. The deterministic baseline (CNP) is presented. We could observe that the performance of the CNP is close to or slightly better than the NP+CNP in Tables 1 of the manuscript. However, the CNP model could lose the functional variability as shown in Figure 2.

## D  ADDITIONAL EXPERIMENTAL DETAILS ON THE IMAGE COMPLETION TASK.

**Setup.** The image completion tasks are performed to validate the performance of the models in more complex function spaces (Garnelo et al., 2018b;a; Kim et al., 2019) Here, we treat the image samples from MNIST (LeCun et al., 1998) and CelebA (Liu et al., 2015) as unknown functions. The task is to predict a mapping from normalized 2D pixel coordinates $x_i$ $(\in [0, 1]^2)$ to pixel intensities $y_i$ $(\in \mathbb{R}^1$ for greyscale, $\in \mathbb{R}^3$ for RGB) given some context points. For 2D regression experiments, we used the same *learned variance* baselines implemented in the GP data regression task, except the input and output of the decoder are changed according to dataset, e.g., $x_i \in [0, 1]^2$, and $y_i \in \mathcal{R}^1$ for MNIST (or $\in \mathcal{R}^3$ and $r^t = 1024$ for CelebA). At each iteration, the images in the training set are split into $S$ context and $N$ target points (e.g., $S \sim U(3, 197)$, $n \sim U[N + 1, 200)$ at train and $S \sim U(3, 197)$, $N = 784 - S$ at validation). Adam optimizer with a learning rate 4e-4 and 16 batches with 300 epochs were used for training. The validation is performed on the separated validation images set. To see the generalization performance on a completely new dataset, we also tested the models trained on MNIST to the Omniglot validation set.

## E  ADDITIONAL EXPERIMENTS ON 1D FUNCTION REGRESSION ON A TOY TRIGONOMETRY DATASET

**Setup.** To further investigate how the proposed meta-model utilizes the common structures of the NN parameters, we trained a small NVDP model (of the size of 13-12-12-2) on a mixture of scaled trigonometric functions: $x$ is sampled in a range of $[-\pi, \pi]$, and $y$ is determined by $y = a * f(2 * x - b * \pi)$ where $f$ is one of three functions sine, cosine, and tanh with the probability of one third, and $a \sim \mathcal{U}(1.5, 2)$ and $b \sim \mathcal{U}(-0.1, 0.1)$. We used the training procedures in the next section except the the simple model architecture and learning rate $5e - 4$.

For the small NVDP model on 1D function regression with Trigonometry dataset, the following architecture was employed:

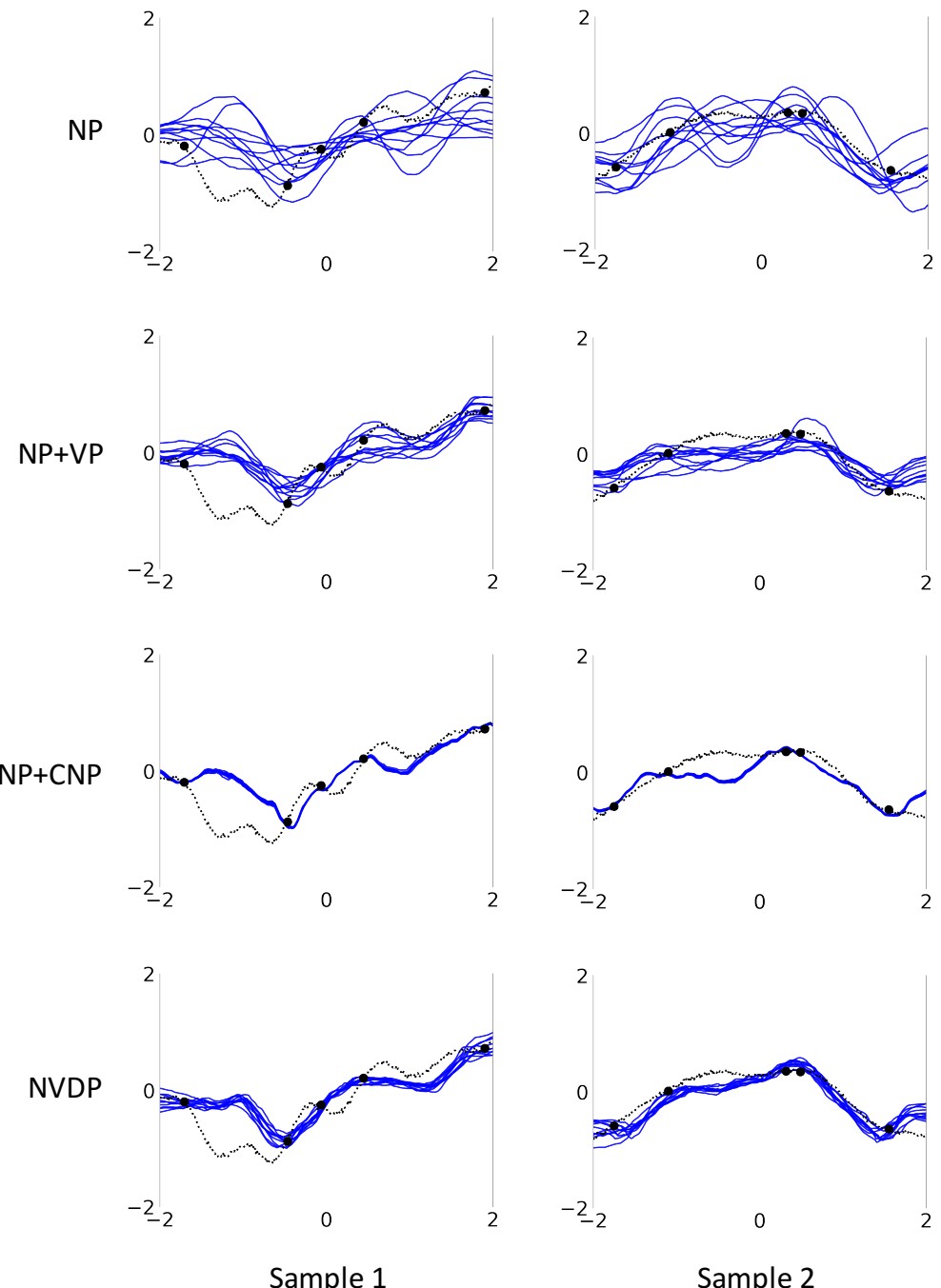

Figure 5: The additional 1D few-shot regression results of the models on GP dataset in *fixed variance* settings. The black dotted lines represent the true unknown task functions. Black dots are a few context points ($S = 5$) given to the posteriors. The blue lines are mean values predicted from the sampled NNs.

**(Deterministic) Feature Encoder.** $r(\mathbf{x}_s^t, \mathbf{y}_s^t) : 2 \times |C| \underbrace{\xrightarrow{\text{lin+relu}} 12 \times |\mathcal{C}|}_{\text{6 times}} \xrightarrow{\text{mean}} 12.$

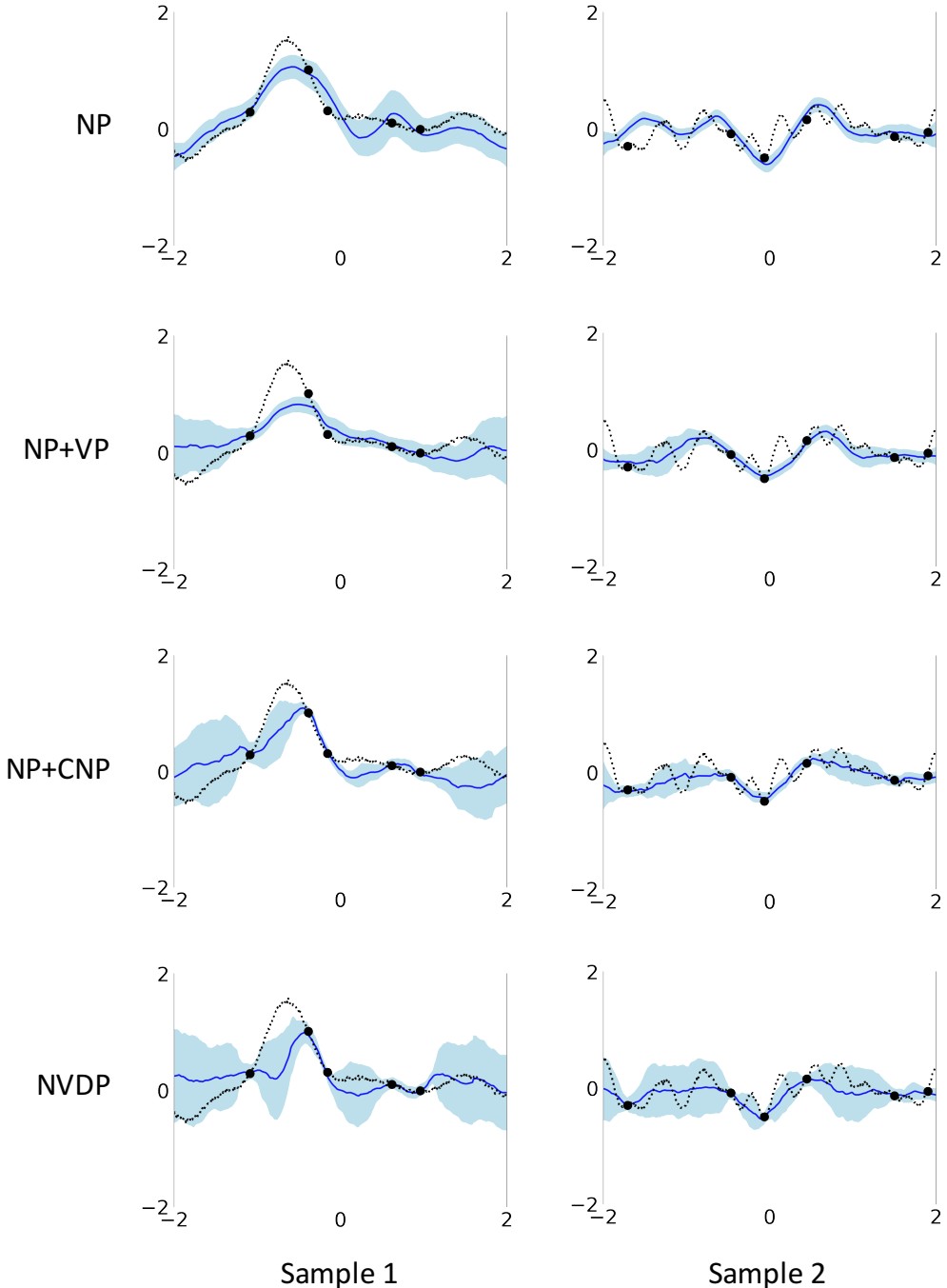

Figure 6: The additional 1D few-shot regression results of the models on GP dataset in *learned variance* settings. The black (dash-line) represent the true unknown task function. Black dots are a few context points ($S = 5$) given to the posteriors. The blue lines and light blue area are mean values and variance predicted from the sampled NNs, respectively.

**Decoder (Agent).** $f(x_i, r) : (1 + d_r) \underbrace{\xrightarrow{\text{lin+relu}} 12}_{\text{2 times}} \xrightarrow{\text{lin+relu}} 2 \xrightarrow{\text{split}} (\mu, \sigma)$ (where $\sigma = 0.1 + 0.9 \cdot$ soft-plus(logstd)).

| Image Dataset | | CNP |
|---|---|---|
| *MNIST* | LL | $0.87(\pm 0.16)$ |
| | RLL | $1.14(\pm 0.08)$ |
| | PLL | $0.83(\pm 0.15)$ |
| *MNIST* | LL | $0.68(\pm 0.11)$ |
| to | RLL | $0.99(\pm 0.08)$ |
| *Omniglot* | PLL | $0.64(\pm 0.13)$ |
| *CelebA* | LL | $0.78(\pm 0.15)$ |
| | RLL | $0.93(\pm 0.05)$ |
| | PLL | $0.77(\pm 0.15)$ |

Table 5: An additional summary of the 2D image completion tasks on the MNIST, CelebA, and Omniglot dataset. The deterministic baseline (CNP) is presented. We could observe that the performance of the CNP is close to or slightly better than the NP+CNP in Tables 2 of the manuscript. However, the CNP model could lose the functional variability as shown in Figure 4.

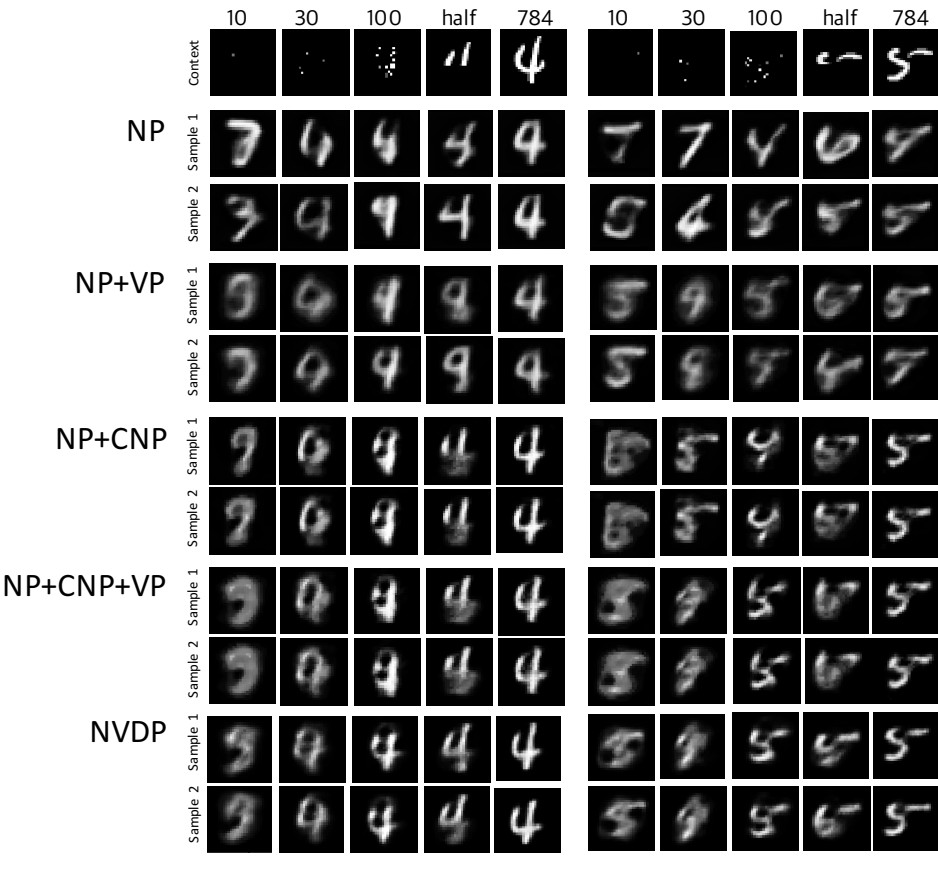

Figure 7: The additional results from the 2D image completion tasks on MNIST dataset. Given the observed context points (10, 30, 100, half, and full pixels), the mean values of two independently sampled functions from the models (i.e. NP, NP+CNP and NVDP (ours)) are presented.

**Meta Model.** $g^{(l)}(r) : d_r \underbrace{\xrightarrow{\text{lin+Mish}} 12}_{4 \text{ times}} \xrightarrow{\text{lin+Mish}} (K^{(l)}, D^{(l)}, 1) \xrightarrow{\text{split}} (\mathbf{a}, \mathbf{b}, \mathbf{c})$

where $g^{(l}(r)$ is the meta NN for the $l$-th layer of the decoder, and $K^{(l)} \times D^{(l)}$ is the number of parameters in the $l$-th layer.

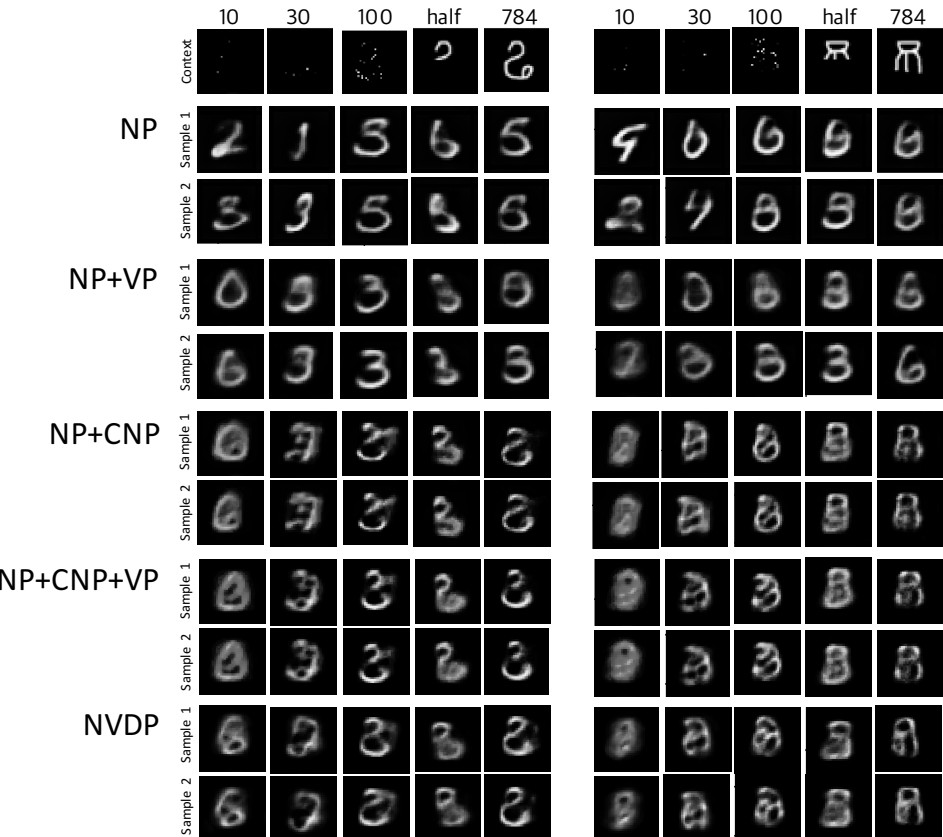

MNIST (train) to Omniglot (test)

Figure 8: The additional results from the 2D image completion tasks on Omniglot dataset. Given the observed context points (10, 30, 100, half, and full pixels), the mean values of two independently sampled functions from the models (i.e. NP, NP+CNP and NVDP (ours)) are presented.

**Results.** Figure 11 displays the trained NVDP on the trigonometry function dataset. The NVDP could capture the probability of dropout for each parameter $\theta$ of the agent, successfully predicting the task-specific trigonometric functions conditioned on the small ($S = 4$) context set. This shows that the task-specific dropout rates can transform a single conventional NN agent to express multiple functions. It is interesting to see that the contexts from sine and cosine functions yield similar dropout rates for the second layer. On the other hand, the contexts from tanh function results in different dropout structures for all layers.

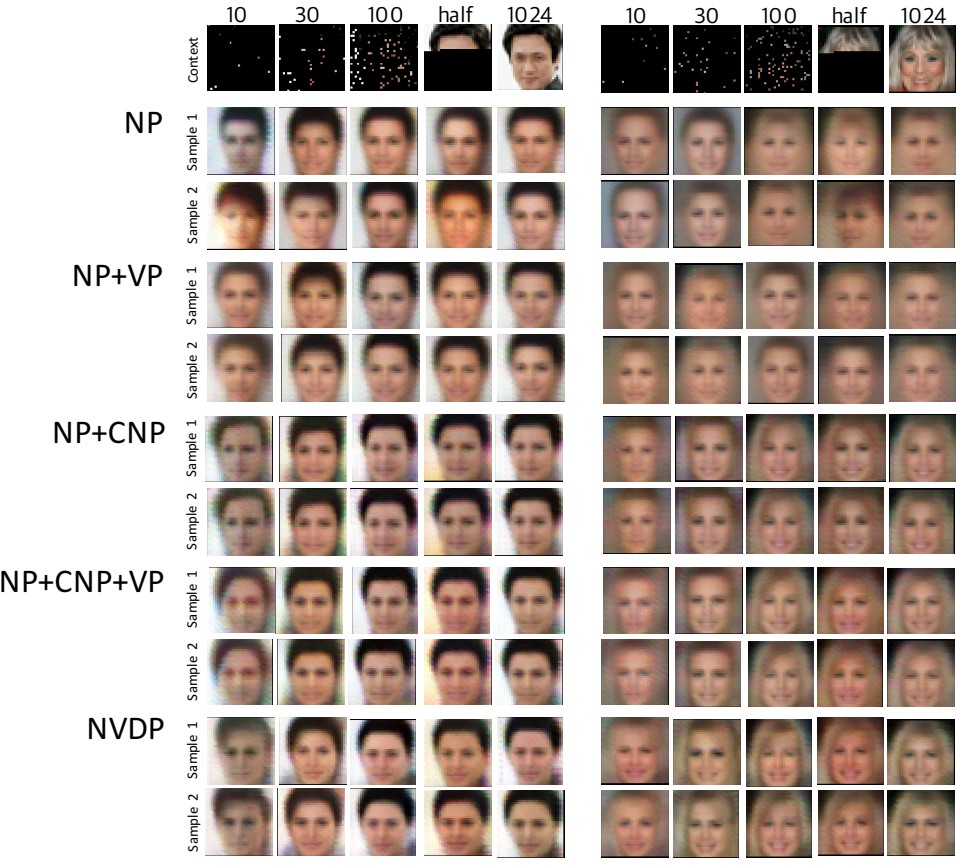

Figure 9: The additional results from the 2D image completion tasks on CelebA dataset. Given the observed context points (10, 30, 100, half, and full pixels), the mean values of two independently sampled functions from the models (i.e. NP, NP+CNP and NVDP (ours)) are presented.

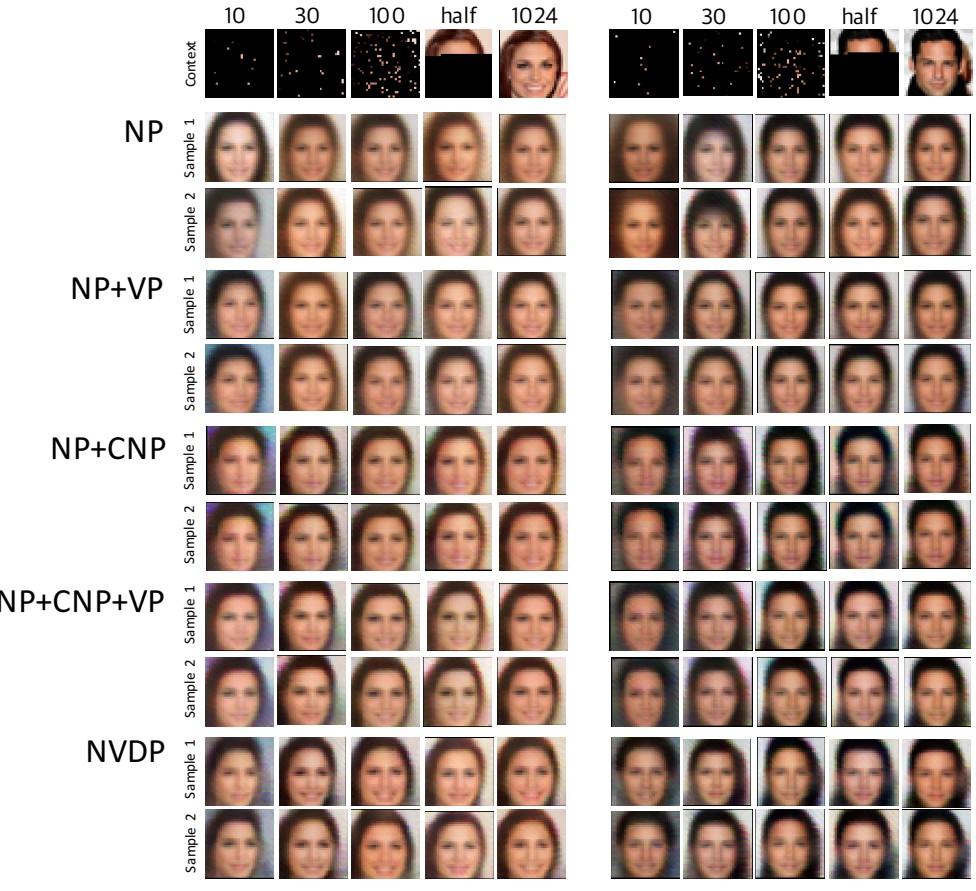

Figure 10: The additional results from the 2D image completion tasks on CelebA dataset. Given the observed context points (10, 30, 100, half, and full pixels), the mean values of two independently sampled functions from the models (i.e. NP, NP+CNP and NVDP (ours)) are presented.

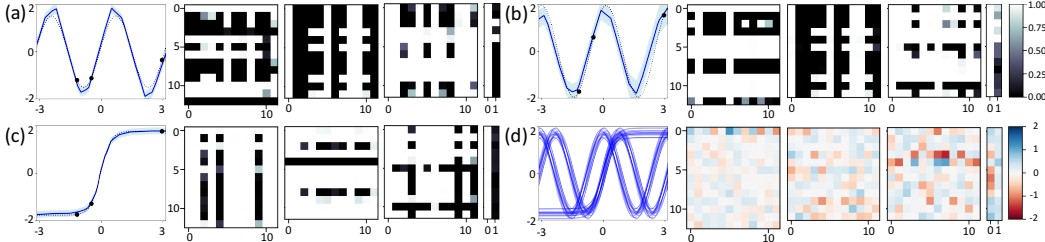

Figure 11: (a) sine, (b) cosine, (c) tanh function (left) with the probabilities of using parameters (1-$\mathbf{P}^t$) (right) predicted with small NVDPs (13-12-12-2) conditioned on 4-shot context points (black dots), and (d) the trigonometry dataset (left) and the deterministic shared NN parameters $\theta$ (right).

