# OpenReview forum: "Neural Variational Dropout Processes"
_ICLR.cc/2022/Conference — ICLR 2022 Poster_

### Official Review · Reviewer_uz6u · 2021-10-31

**Correctness:** 4
**Technical Novelty And Significance:** 3
**Empirical Novelty And Significance:** 2
**Recommendation:** 6
**Confidence:** 3

**Main Review:**

Some strengths:
1. The newly proposed NVDPs are more robust to the model-collapsing or overfitting issues, comparing with existing approaches.
2. The proposed variational prior can regularize the task-specific dropout rates in the optimization of the conditional dropout posterior.
3. Comparing with previous related approaches, the complexity is greatly reduced, making the algorithm more applicable to larger data set with scalability.

Some weaknesses:
1. This approach still falls into the SGVB framework when coming to the optimization of the variational parameters, using the re-parameterization trick. This limits its novelty, and restrains the potential of this framework to investigate more complicated distributions other than Bernoulli.
2. Recently, meta learning has been popular in some application domains, e.g., NLP. The experiments are majorly focused on highly optimized data sets, weakening its wider application on more practical tasks and usage in real scenarios.


**Summary Of The Paper:**

The authors present an interesting model-based meta-learning approach, Neural Variational Dropout Processes (NVDP), by constructing a conditional dropout posterior to predict task-specific dropout rates of model parameters, conditioned on the observations. Together with NVDP, the authors also propose a new type of prior, the task-specific variational prior, approximated by the same posterior model conditioned on the whole task data, which supports robust training. They show the NVDPs improve the model's adaptation, functional variability and generalization to new tasks empirically in the experiments.

**Summary Of The Review:**

I recommend this paper to be accepted, as its novelty on the newly developed variational prior differs from existing works. This is based on the hypothesis that the conditional posterior model to approximate the optimal prior shall depend on the whole data set. This claim is supported by the assumption that the variational model conditioned on the aggregated set with larger context shall be closer to the optimal task-specific prior, than only conditioned on a subset, which is a specific task. In the experiments, the variational prior shows a robust regularization effect.

---

> ### Author Response · Authors · 2021-11-15
> **Response to Reviewer uz6u**
>
> **Thank you for encouraging feedback and constructive comments. Below we reply to your supportive comments.**
>
> **Q1**. This approach still falls into the SGVB framework when coming to the optimization of the variational parameters, using the re-parameterization trick. This limits its novelty and restrains the potential of this framework to investigate more complicated distributions other than Bernoulli.
>
> **A1**. We would like to touch on some differences in our works from the existing SGVB approaches. First, this work is an initial attempt to connect the exchangeable set representation [1,2,3] to approximate the Bernoulli dropout posterior. Second, we also suggest a new concept of variational (reversed) prior in the SGVB framework. In addition, our approach employs the explicit decomposition between the deterministic shared parameters of the likelihood model and the parameters of the meta-model in the variational inference, unlike other related works such as NPs or VERSA. We expect this property allows our model can join with different state-of-the-art optimization-based meta-learning approaches naturally. Lastly, we believe the product of the Bernoulli model, as it has a simplistic form, can also readily be extended to much-complicated distributions or architectures. For example, we expect it would be possible to utilize the NVDPs with the CNN architectures [4] or the structured dropout approach [5].
>
> **Q2**. Recently, meta-learning has been popular in some application domains, e.g., NLP. The experiments are majorly focused on highly optimized data sets, weakening their wider application on more practical tasks and usage in real scenarios.
>
> **A2**. We will try to expand the application area to the more diverse domain as the future works. Thank you for your valuable time for the review.
>
> **References**
>
> [1] Harrison Edwards and Amos Storkey. Towards A Neural Statistician. In ICLR, 2017.
>
> [2] Juho Lee, Yoonho Lee, Jungtaek Kim, Adam R. Kosiorek, Seungjin Choi, Yee Whye Teh. Set Transformer. In PMLR, 2019
>
> [3] Benjamin Bloem-Reddy and Yee Whye Teh. Probabilistic symmetry and invariant neural networks. In JMLR, 2020.
>
> [4] Jonathan Gordon, Wessel P. Bruinsma, Andrew Y. K. Foong, James Requeima, Yann Dubois, Richard E. Turner. Convolutional Conditional Neural Processes. In ICLR, 2020.
>
> [5] Son Nguyen, Duong Nguyen, Khai Nguyen, Khoat Than, Hung Bui, Nhat Ho. Structured Dropout Variational Inference for Bayesian Neural Networks. In NeurIPS, 2021.

---

### Official Review · Reviewer_HrLQ · 2021-11-02

**Correctness:** 3
**Technical Novelty And Significance:** 3
**Empirical Novelty And Significance:** 2
**Recommendation:** 8
**Confidence:** 4

**Main Review:**

## Summary
The authors propose Neural Variational Dropout Processes (NVDPs), a novel approach to meta-learning. First, the task of meta-learning is interpreted as performing Bayesian inference of the model parameters under different environments (multi-task learning). For training, the authors perform approximate Bayesian inference by optimizing a variational lower bound on the multi-task likelihood (eq. (1)). Two neural networks are trained: a base neural network with deterministic parameters that is used across tasks, and a meta neural network that outputs task-specific dropout rates for the weights of the base network given a small support set of data points S (amortized posterior). The authors link their approach to variational dropout (Kingma et al., 2015), but utilize Bernoulli dropout instead of a Gaussian approximation. For regularization the authors use a 'variational prior' that is essentially the aggregated posterior over all data points in a task. This formulation of approximate posterior and variational prior allows the authors to derive a KL-term that is independent of the base network's deterministic parameters (appendix B).

The authors place their work in the context of model-based approximate posterior approaches (Gordon et al., 2018; Garnelo et al., 2018a;b; Kim et al., 2019; Iakovleva et al., 2020) which all utilize a formulation akin to (eq. 1), with a semantic differences in the outputs of the approximate posterior. The authors claim that their formulation overcomes the scalability problems of VERSA (Gordon et al., 2018), and avoids posterior collapse as is sometimes observed in Neural Processes (NPs) (Garnelo et al., 2018a;b).

The proposed method is evaluated on several regression and classifications problems:

1. A 1d regression problem where tasks are sampled from a Gaussian Process
2. Active learning on the above task
3. Image completion on CelebA, MNIST, and Omniglot (only test)
4. Few-shot classification on Omniglot and MiniImageNet

In all problems domains, the results suggest either state-of-the-art or compentitive performance of NVDPs as compared to other approaches.

## Opinion & Recommendation
I am in support of accepting this paper. The paper makes a nice connection between meta-learning and variational dropout, resulting in an overall elegant approach. The use of dropout for adaptation of a model to different tasks is (as far as I am aware) novel, which would already justify publication. NVDPs are closely related to Neural Processes (NPs). However NPs can suffer from the fact that the outputs of the approximate posterior are ignored by the downstream decoder model. This is not the case for NVDPs as they directly modify the decoder through the dropout process.

The presented results give a mostly positive impression of the approach. Judging from the image completion problems, however, the limitations of all compared approaches (including NVDPs) becomes apparent. Given a fully sampled support set the resulting image reconstructions are still very blurry. See Kim et al. (2019, fig. 4a) for much more crisp reconstructions. The author's claim "On the other hand, the samples from NVDPs exhibit sharp reconstruction results while also showing a reasonable amount of variability." is in my option an overstatement.

## Questions
- The authors mention at the end of section 3 that the local reparametrization trick is used for variance reduction. But it looks like it is still necessary loop over different tasks *t*. Is this true? If so, could you discuss the computational limitations of this for large *T* ?
- Eq. (3) gives a low rank approximation of the dropout rate *P*. It is a product of 3 Bernoulli probabilities. Could you make some comments about the numerical stability of this approximation?

**Summary Of The Paper:**

I will put a few bullet points here. For a more detailed summary, see below.
- Novel meta-learning approach
- Utilises dropout to adapt a base network to new tasks
- Amortised variational inference is used to train meta model and base model jointly
- The meta model outputs task specific dropout rates
- This results in a conditional dropout posterior akin to variational dropout
- A *variational prior* over task-dependent dropout rates allows simple calculation of the KL-divergence
- The approach is evaluated in 4 different problem setting. SOTA or competitive in most.

**Summary Of The Review:**

I recommend acceptance of the presented paper. A novel idea, combined with a good practical justification, and strong results make this paper worth for publication.

---

> ### Author Response · Authors · 2021-11-15
> **Response to Reviewer HrLQ**
>
> **Thank you for your detailed review and encouragement of our paper. Below we reply to your supportive comments.**
>
> **Q1.** The authors mention at the end of section 3 that the local reparametrization trick is used for variance reduction. But it looks like it is still necessary to loop over different tasks t. Is this true? If so, could you discuss the computational limitations of this for large $T$?
>
> **A1.** Your understanding is correct. For example, the total number of tasks $T=1200$ in the Omniglot training dataset and $T=202599$ in the CelebA dataset. The actual training is performed with $T'$ mini-batch of randomly sampled tasks per  epoch (with M random data points sampled for each task). We use $T’=16$ in all the regression tasks, $T’=5$ or $20$ in classification tasks. The computational restriction is that the overall training time can increase linearly due to the large $T$. We will update equation (6) and the experiment section to clarify this.
>
> **Q2.** Eq. (3) gives a low-rank approximation of the dropout rate P. It is a product of 3 Bernoulli probabilities. Could you make some comments about the numerical stability of this approximation?
>
> **A2.** Yes. We tested many different methods for the low-rank approximation form using the naive logit sum or product with other variants of activation functions. We found that the product of three Bernoulli probabilities shows the most stable result to the various initial seeds or the NN weights. The product of three probabilities is naturally normalized as in the range of $(0,1)$. We did not experience any particular gradient vanishing or exploding problem due to the product computation. We also think this numerical stability is partially due to the sigmoid with a learnable temperature parameter $\tau$, as mentioned in the manuscript. The softmax is applied on the scaled logits (i.e., logits/$\tau$). Thus, the temperature can control the softness of softmax in predicting a probability distribution over classes. In practice, we restricted the learnable $\tau$ range between $(0.5, 5)$ for all the experiments. One possible converge issue of using the variational prior is that dropout rates can converge to zeros during the early training period due to more strong gradients signal from the KL than from the likelihood. We could resolve this local optimum behavior by clipping the dropout rate range in $(0.01, 0.99)$ as mentioned in appendix B.
>
> **Q3.** The presented results give a mostly positive impression of the approach. Judging from the image completion problems, however, the limitations of all compared approaches (including NVDPs) becomes apparent. Given a fully sampled support set the resulting image reconstructions are still very blurry. See Kim et al. (2019, fig. 4a) for much more crisp reconstructions.
>
> **A3.** Although we have strictly followed the same architectural description presented in [1], the baselines seem to have slightly blurry results than the original papers. We conjecture this might be due to the activation of the output layer or some other architectural details. We will further investigate this and tone down the possible overstatement in the experimental description. Nonetheless, we believe the overall performance tendency of NVDP will not be changed much (since baselines and NVDP shares similar architecture). Thank you for your valuable time for the review.
>
> **References**
>
> [1] Hyunjik Kim, Andriy Mnih, Jonathan Schwarz, Marta Garnelo, Ali Eslami, Dan Rosenbaum, Oriol Vinyals, Yee Whye Teh. Attentive Neural Processes. In ICLR, 2019.

---

### Official Review · Reviewer_8oez · 2021-11-04

**Correctness:** 3
**Technical Novelty And Significance:** 3
**Empirical Novelty And Significance:** 3
**Recommendation:** 6
**Confidence:** 3

**Main Review:**

Strength:
+ The paper is clearly written and easy to follow.
+ The evaluation of NP-based methods using Reconstructive Log-likelihood (RLL) and predictive log-likelihood (PLL) is an interesting idea.
+ The reasoning of the novel prior is clear and understandable.
+ The experiments are comprehensive.

Weakness:
- Predictive accuracy (Table 3) alone is not sufficient to evaluate the strength of the uncertainty-aware few shot classification model, i.e. many recent few-shot models can easily perform significantly better than the results presented here (e.g. Meta-Learning with Differentiable Convex Optimization).
- Experiments to demonstrate the applicability of uncertainty estimates (e.g., an active learning setting similar to Bayesian MAML) will further strengthen the results.
- The comparison with CNP models alone is missing. The evaluation is done with NP + CNP instead.
- Comparison with state-of-the-arts NP works will help better evaluate the model’s performance, for example, "Convolutional conditional neural processes, Gordon et al. ICLR 2020", "Meta-learning stationary stochastic process prediction with convolutional neural processes,
Foong et al, NeurIPS 2020", or "Attentive neural processes, Kim et al, ICLR 2019".
- Although the authors claim to address the issues such as mode-collapsing or over-fitting behavior in the context of meta-learning, the results to support the claim are missing in the paper.



**Summary Of The Paper:**

The paper proposes a novel Bayesian meta-learning approach to model conditional posterior distribution via an amortized variational inference framework based on task-specific dropout. For tasks with few learning examples, the paper discusses the a novel low-rank product of Bernoulli expert meta-models to efficiently obtain full conditional posterior model over the neural network’s parameters. Few-shot learning experiments from Gaussian Process samples show that the model achieves better performance. The proposed method also outperforms existing models on active learning task with better uncertainty calibration. The method also shows better performance on image completion and few-shot classification tasks.

**Summary Of The Review:**

an interesting idea, but there is room to further strengthen the paper, particularly for the experiments.

---

> ### Author Response · Authors · 2021-11-15
> **Response to Reviewer 8oez (2 of 2)**
>
> **Q4.** The comparison with CNP models alone is missing.
>
> **A4.** We will update the following results of CNP in the appendix.
>
> | (GP)   |      Fixed variance        |  Learned Variance |
> |----------|:-------------:|:-----------------------------:|
> | LL    | -0.94 ($\pm$0.05)          |  0.72 ($\pm$ 0.54) |
> | RLL |   -0.93 ($\pm$ 0.01)    |  1.03 ($\pm$ 0.38)  |
> | PLL | -0.94 ($\pm$ 0.05)      |     0.69 ($\pm$ 0.53)  |
>
>
> |  (2D)  |     MNIST  |     MNIST to Omniglot  |     CelebA  |
> |----------|:-------------:|:-----------------------------:|:-----------------------:|
> |     LL  | 0.87 ($\pm$ 0.16)   |    0.68 ($\pm$0.11)       |   0.78 ($\pm$0.15)   |
> |    RLL | 1.14 ($\pm$ 0.08)   |    0.99 ($\pm$0.08)       |   0.93 ($\pm$0.05)   |
> |    PLL | 0.83 ($\pm$ 0.15)   |     0.64 ($\pm$0.13)      |    0.77 ($\pm$0.15)  |
>
> **Q5.** The evaluation is done with NP + CNP instead. Although the authors claim to address the issues such as mode-collapsing or over-fitting behavior in the context of meta-learning, the results to support the claim are missing in the paper.
>
> **A5.** In the regression tasks (Figure 2 and Figure 4), we presented the qualitative result of the NP+CNP, instead of the CNP model, to highlight the posterior-collapsing behavior of NP+CNP. This happens due to the collapsing behavior of task-specific latent variables; when the deterministic variable is added, the likelihood model tends to ignore the task-specific latent variable. In addition, we could observe that the performance of the CNP is close to or slightly better than the NP+CNP in Tables 1 and 2 of the manuscript and in (A4). In other words, if we utilize the deterministic path in the NP approaches (CNP or NP+CNP), these models can lose epistemic variability in some tasks. On the other hand, the NVDP accomplishes both good functional variabilities and generalization performance on new unseen tasks in various tasks. We will update the claim and supportive factors in the manuscript more clearly. Thank you for your valuable time for the review.
>
> **References**
>
> [1] Kwonjoon Lee, Subhransu Maji, Avinash Ravichandran, Stefano Soatto. Meta-Learning with Differentiable Convex Optimization. In CVPR, 2019.
>
> [2] https://opencv.org/understanding-transductive-few-shot-learning/
>
> [3] Wei-Yu Chen, Yen-Cheng Liu, Zsolt Kira, Yu-Chiang Frank Wang, Jia-Bin Huang. A Closer Look at Few-shot Classification. In ICLR, 2019.
>
> [4] Chelsea Finn, Kelvin Xu, Sergey Levine. Probabilistic Model-Agnostic Meta-Learning. In NeurIPS, 2018.
>
> [5] Jaesik Yoon, Taesup Kim, Ousmane Dia, Sungwoong Kim, Yoshua Bengio, Sungjin Ahn. Bayesian Model-Agnostic Meta-Learning. In NeurIPS, 2018.
>
> [6] Hyunjik Kim, Andriy Mnih, Jonathan Schwarz, Marta Garnelo, Ali Eslami, Dan Rosenbaum, Oriol Vinyals, Yee Whye Teh. Attentive Neural Processes. In ICLR, 2019.
>
> [7] Jonathan Gordon, Wessel P. Bruinsma, Andrew Y. K. Foong, James Requeima, Yann Dubois, Richard E. Turner. Convolutional Conditional Neural Processes. In ICLR, 2020.
>
> [8] Andrew Y. K. Foong, Wessel Bruinsma, Jonathan Gordon, Yann Dubois, James Requeima, Richard E. Turner. Meta-Learning Stationary Stochastic Process Prediction with Convolutional Neural Processes. In NeurIPS, 2020.

---

> ### Author Response · Authors · 2021-11-15
> **Response to Reviewer 8oez (1 of 2)**
>
> **Thank you for your valuable review and positive feedback on our paper. Below we reply to your supportive comments.**
>
> **Q1.** Predictive accuracy (Table 3) alone is not sufficient to evaluate the strength of the uncertainty-aware few-shot classification model, i.e. many recent few-shot models can easily perform significantly better than the results presented here (e.g. Meta-Learning with Differentiable Convex Optimization).
>
> **A1.** Thank you for mentioning the inspiring paper [1]. We agree with your opinion. We are also aware of some other recent deterministic approaches show excellent prediction accuracy in the few-shot classification tasks [1,2]. However, the main contribution of our paper was to propose a new conditional dropout posterior model and variational (reversed) prior compared with other model-based Meta-learning approaches. Thus, we assigned the standard feature extractor architecture of the related methods, such as the fully connected NN of NPs in the regression and ConvNet of VERSA in the classification tasks for a similar comparison. We believe the deeper feature extractors such as WRN or ResNet are another important performance factor as discussed in [3]. Incorporating these deeper feature extracting architectures into NVDPs is interesting future work.
>
> **Q2.** Experiments to demonstrate the applicability of uncertainty estimates (e.g., an active learning setting similar to Bayesian MAML) will further strengthen the results.
>
> **A2.** In the manuscript, the active learning experiment (e.g., a setting similar to Probabilistic MAML [4]) was already performed in few-shot regression tasks but could not be in classification. We will try to update the active learning setting of [5] in the appendix.
>
> **Q3.** Comparison with state-of-the-arts NP works will help better evaluate the model’s performance, for example, "Convolutional conditional neural processes, Gordon et al. ICLR 2020", "Meta-learning stationary stochastic process prediction with convolutional neural processes, Foong et al., NeurIPS 2020", or "Attentive neural processes, Kim et al., ICLR 2019".
>
> **A3.** Thank you for mentioning the novel works [6,7,8]. The related works will be mentioned in the manuscript. In fact, we had already examined the exchangeable representation from the attention model proposed by Kim et al. [6] to the conditional dropout posterior in NVDPs and observed some notable performance gain in the regression tasks. However, we did not report those results since we thought the improvement was somewhat orthogonal to our main contributions and experimental scope. We believe the extension of the NVDPs with the CNN [7,8] and attention mechanism is also possible and very interesting future work.

---

### Author Response · Authors · 2021-11-23
**To all reviewers**

We thank all reviewers for their supportive feedback on this work.

During the discussion period, we updated the paper to reflect the opinions of reviewers.
Also, many typos and errors were corrected.

The major changes can be summarized as follows.
 - Updated equation 6 and the description of the mini-batch Monte-Carlo estimator in the paper.
 - Reduced possible overestimation in our claims or experiments.
 - Organized related works mentioned in the discussion.
 - Updated the experiment results of the CNP model in the appendix.

We hope our responses and updates can provide further insight to many readers.
We are currently working on other experiments and will include them in the appendix when these are ready. We again thank all reviewers.

---

### Public Comment · ~Zhibin_Duan1 · 2022-02-28
**An interesing work**

Congratulations! This is a very interesting work, and can you release code for my further study, thank you very much.

---

> ### Public Comment · ~Insu_Jeon1 · 2022-02-28
> **Thank you**
>
> We appreciate your interest in our work. We plan to release the code in public near or right after the conference as the refactoring and final preparation are done. Please stay tuned for our update.

---

### Public Comment · ~Insu_Jeon1 · 2022-08-16
**Our code for reproducing the experiment is available**

Here is a link (https://github.com/insuj3on/NVDPs) to the code for reproducing the experiment presented in the paper.
Thank you for your interest.

---

### Decision · Program_Chairs · 2022-01-20

**Decision:**

Accept (Poster)

**Comment:**

This paper proposes a novel model-based Bayesian meta-learning approach that combines a novel conditional dropout posterior a new variational prior for the data-efficient learning and adaptation of deep neural networks. It is applied to tasks such as 1D stochastic regression, image inpainting, and classification.

Overall, this paper received positive reviews: Reviewers thought that the "the paper makes a nice connection between meta-learning and variational dropout, resulting in an overall elegant approach" and that the "reasoning of the novel prior is clear and understandable" while the "experiments are comprehensive".

Given the agreement of the reviewers and the novel use of dropout for adaptation of a model to different tasks, I recommend accepting this paper.